# Safe Reinforcement Learning with Preference-based Constraint Inference

**Chenglin Li** [1]   **Grant Ruan** [2]   **Hua Geng** [1]

## Abstract

Safe reinforcement learning (RL) is a standard paradigm for safety-critical decision making. However, real-world safety constraints can be complex, subjective, and even hard to explicitly specify. Existing works on constraint inference rely on restrictive assumptions or extensive expert demonstrations, which are not realistic in many real-world applications. How to cheaply and reliably learn these constraints is the major challenge we focus on in this study. While inferring constraints from human preferences offers a data-efficient alternative, we identify popular Bradley-Terry (BT) models fail to capture the asymmetric, heavy-tailed nature of safety costs, resulting in risk underestimation. It is still rare in the literature to understand the impacts of BT models on the downstream policy learning. To address the above knowledge gaps, we propose a novel approach namely Preference-based Constrained Reinforcement Learning (PbCRL). We introduce a novel dead zone mechanism into preference modeling and theoretically prove that it encourages heavy-tailed cost distributions, thereby achieving better constraint alignment. Additionally, we incorporate a Signal-to-Noise Ratio (SNR) loss to encourage exploration by cost variances, which is found to benefit policy learning. Further, two-stage training strategy is deployed to lower online labeling burdens while adaptively enhancing constraint satisfaction. Empirical results demonstrate that PbCRL achieves superior alignment with true safety requirements and outperforms state-of-the-art baselines in terms of safety and reward. Our work explores a promising and effective way for constraint inference in Safe RL, with great potential in various safety-critical applications.

[1]Department of Automation, Tsinghua University, Beijing, China [2]Laboratory for Information & Decision Systems, Massachusetts Institute of Technology, Cambridge, MA, USA. Correspondence to: Hua Geng <genghua@tsinghua.edu.cn>.

*Proceedings of the $43^{rd}$ International Conference on Machine Learning*, Seoul, South Korea. PMLR 306, 2026. Copyright 2026 by the author(s).

## 1. Introduction

A common paradigm to integrate safety considerations in sequential decision making is through Safe Reinforcement Learning (RL) (Garcıa & Fernández, 2015; Gu et al., 2024b). Typically formulated as a Constrained Markov Decision Process (CMDP) (Altman, 2021), Safe RL aims to maximize reward while meeting safety constraints, often expressed as bounds on the expectation of cumulative cost.

Apart from those explicit constraints such as univariate bounds, there is a large family of implicit constraints in the real world that could be highly complicated, trajectory-based, and perhaps seldom well-specified. Many of these constraints even involve human preferences and reflect trade-offs between different criteria. In robotic navigation or autonomous driving, for example, unsafe policies causing potential collisions, uncomfortable maneuvers, and other risky outcomes are never easy to quantify or subjective to human preference (Cosner et al., 2022). Unfortunately, this is not an isolated case; even with dedicated system engineering, it is still too expensive and impractical to construct proper constraints that accurately reflect all safety requirements (Xu & Liu, 2024).

Constraint inference from data has emerged as a promising alternative to manual specification. Existing approaches include Inverse Reinforcement Learning (IRL) (Ng et al., 2000; Scobee & Sastry, 2020), Control Barrier Functions (CBF) (Robey et al., 2020) and robust optimization (Xu & Liu, 2024). These methods typically rely on extensive expert demonstrations to infer the underlying constraints. However, collecting such fine-grained demonstrations from experts is often expensive, time-consuming, or even infeasible in certain scenarios (Arora & Doshi, 2021).

A more data-efficient alternative is learning through human feedback of comparisons instead of demonstrations (Christiano et al., 2017; Wu et al., 2022). This is a typical human-in-the-loop data collection, because options are provided by the RL agent and human only need to compare and then select. Learning from these comparison datasets is not straightforward and requires redesign of the learning algorithms. In the literature, prior works often restrict the constraint function class (e.g. linear, continuous) (Sui et al., 2015; Amani et al., 2021; Wachi & Sui, 2020) or require dense state-level feedback (Bennett et al., 2023). These

settings simplify the problem but limit the applicability to more complex scenarios as well.

A few recent works have borrowed the Bradley-Terry (BT) model (Bradley & Terry, 1952) from preference learning to infer constraints from human preference (Reddy Chirra et al., 2024; Dai et al., 2024). Typically, a binary cost function is learned in (Reddy Chirra et al., 2024) from human feedback with various granularity-at trajectory level in simple tasks or state level in complex ones. Within the domain of language model alignment, works such as (Dai et al., 2024) also leverage the BT model to constrain potentially harmful behaviors in generated text responses. Details of related works are in Appendix A. Compared to densely annotated demonstrations, preference data, collected usually at trajectory level, is more coarse-grained and cheaper to obtain. This makes preference learning a more practical and data-efficient solution for constraint inference.

However, we identify two major drawbacks when directly applying standard BT models for constraint inference in Safe RL. First, there is a critical misalignment between the ranking objective of BT models and the distributional requirements of Safe RL. Standard BT models are inherently designed for learning the cost rank (relative order) between trajectories, but can be biased in estimating the true distributions (need absolute values) (Bradley & Terry, 1952). Since Safe RL relies on the expected cumulative cost to formulate constraints (Altman, 2021), two different cost distributions with identical rankings may lead to different expectation values, as well as distinct constraint satisfaction outcomes. Crucially, it is found that BT models tend to infer symmetry but most true cost distributions in Safe RL are heavy-tailed (Yang et al., 2021; Li et al., 2025). This phenomenon arises from the cascading effects of unsafe events: an initial violation can trigger a sequence of subsequent failures. For instance, once a navigation robot collides with an obstacle, it is more likely to be in hazardous situations in the near future, causing the cumulative cost to grow much larger. This cascading effect results in a heavy-tailed distribution of cumulative costs over trajectories. As conceptually illustrated in Figure 1, the mismatch between the symmetric BT-inferred cost distribution and the true heavy-tailed distribution leads to underestimation of expected costs, which will result in aggressive/dangerous policies.

Second, existing methods largely overlook the impact of the learned cost model on downstream policy optimization. Currently, most studies evaluate cost models solely based on prediction accuracy, failing to assess their influence on policy quality (Lambert et al., 2025; Wen et al., 2025). It has been found that a cost model that produces flattened outputs can hinder effective policy updates (Razin et al., 2025). Alternative designs that encourage exploration and cost diversification should be preferred to provide informative signals for policy learning.

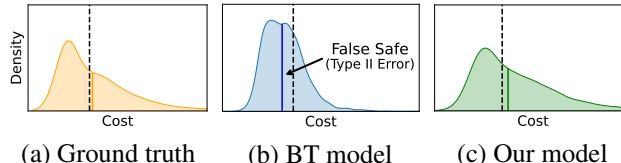

*Figure 1.* Cost distributions: (a) Ground truth: heavy-tailed with expectation (solid line) exceeding safety threshold (dashed line), indicating unsafe. (b) BT model: symmetric with underestimated expectation below the threshold, induces a **Type II Error** where the model falsely believes the constraint is satisfied. (c) Our model: similar heavy-tailed distribution, better approximate expectation of ground truth cost.

To address the above issues, we propose a novel Safe RL approach with preference-based constraint inference, namely Preference-based Constrained Reinforcement Learning (PbCRL). Our contributions are summarized as follows:

1. We develop a novel constraint inference model with dead zone designs. We theoretically prove that the dead zone encourages heavy-tailed cost distributions, which mitigates the constraint underestimation (unsafe and dangerous) with better constraint alignment.

2. A Signal-to-Noise Ratio (SNR) loss is proposed to induce sufficient level of cost variation. This can be regarded as a regularization term to encourage informative signals for effective downstream policy learning.

3. We establish a two-stage training strategy to lower the burden of online labeling and help improve constraint satisfaction. Furthermore, we provide a theoretical convergence guarantee for the overall PbCRL algorithm.

4. We conduct extensive experiments on Safe RL benchmarks. Results demonstrate that PbCRL achieves superior alignment with true safety requirements and outperforms baselines in terms of safety and reward.

## 2. Problem Formulation

### 2.1. Safe RL with Unknown Constraints

Safe RL is formulated on the constrained Markov decision process (CMDP) $\langle S, A, P, r, c, d, \gamma \rangle$, where $S$ denotes the state space, $A$ denotes the action space, $P(\cdot|s, a)$ is the state transition probability function, $r(s, a)$ is the reward function, $c(s, a)$ is the cost function, $d$ is a given safety threshold, and $\gamma \in (0, 1)$ is the discount factor.

The agent interacts with the environment at each time step $t$ by observing the current state $s_t \in S$ and selecting an action $a_t \in A$ from its policy $\pi(\cdot|s)$. Then the agent receives a reward $r(s_t, a_t)$ and a cost $c(s_t, a_t)$ from the environment. The next state $s_{t+1}$ is generated by $P(\cdot|s_t, a_t)$. Given an initial state $s_0$, the cumulative reward is defined as

$R(s_0, \pi) = \sum_{t=0}^{\infty} \gamma^t r(s_t, a_t)$, and the cumulative cost as $C(s_0, \pi) = \sum_{t=0}^{\infty} \gamma^t c(s_t, a_t)$.

With the above setup, Safe RL is established to learn a policy that maximizes the expected cumulative reward $\mathcal{J}^R(\pi)$ while meeting safety constraints, which is usually expressed as the expected cumulative cost $\mathcal{J}^C(\pi)$ being below a certain threshold $d$. Formally,

$$\max_{\pi} \quad \mathcal{J}^R(\pi) := \mathbb{E}_\pi[R(s, \pi)] = \mathbb{E}_\pi[\sum_t \gamma^t r(s_t, a_t)] \tag{1a}$$

$$\text{s.t.} \quad \mathcal{J}^C(\pi) := \mathbb{E}_\pi[C(s, \pi)] = \mathbb{E}_\pi[\sum_t \gamma^t c(s_t, a_t)] \leq d \tag{1b}$$

In this paper, we consider the situation where the safety constraint is not explicitly defined and must be learned. The constraint includes both the cost function $c(s, a)$ and the threshold $d$. Therefore, we reformulate the safe RL problem in Eq. 1 with inferred constraint:

$$\max_{\pi} \mathcal{J}^R(\pi) = \mathbb{E}_\pi[\sum_t \gamma^t r(s_t, a_t)], \tag{2a}$$

$$\text{s.t.} \ \mathcal{J}^{\hat{C}}(\pi) = \mathbb{E}_\pi[\sum_t \gamma^t \hat{c}(s_t, a_t)] \leq 0 \tag{2b}$$

where Eq. 2b is the constraint with learned cost function $\hat{c}(s, a)$. The threshold is assigned to be 0 without loss of generality. The constraint in Eq. 2b should be functionally equivalent to the true unknown constraint in Eq. 1b, reflecting the underlying safety requirements.

### 2.2. Learning Constraints from Preference

Preference learning aims to learn a utility function from human preference (Christiano et al., 2017; Lee et al., 2021). Feedback is collected in the form of pairwise preferences over two trajectories $\tau_1, \tau_2$, where $\tau$ denotes a sequence of state-action pairs $\{(s_0, a_0), (s_1, a_1), \ldots, (s_T, a_T)\}$ of length $T$. Human annotators provide preference label $(\mu_1, \mu_2)$, where $\mu_1 > \mu_2$ indicates that $\tau_1$ is preferred over $\tau_2$ (denoted as $\tau_1 \succ \tau_2$).

The standard framework assumes human preferences follow a Bradley-Terry (BT) model based on an implicit utility function. In our context, we aim to learn a cost function $\hat{c}(s, a)$ where a lower cost implies a higher preference. Consequently, the probability of preferring $\tau_1$ over $\tau_2$ is modeled as:

$$\hat{\mathbb{P}}(\tau_1 \succ \tau_2) = \sigma(\hat{C}(\tau_2) - \hat{C}(\tau_1)) = \frac{e^{\hat{C}(\tau_2)}}{e^{\hat{C}(\tau_1)} + e^{\hat{C}(\tau_2)}} \tag{3}$$

where $\hat{C}(\tau) = \sum_{(s_t, a_t) \sim \tau} \gamma^t \hat{c}(s_t, a_t)$ denotes trajectory cumulative cost, $\sigma(\cdot)$ is sigmoid function. The cost is optimized by minimizing the pairwise cross-entropy loss on a preference dataset $\mathcal{D}$:

$$\mathcal{L}_{pair} = -\mathbb{E}_\mathcal{D}\Big[\mu_1 \log \hat{\mathbb{P}}(\tau_1 \succ \tau_2) + \mu_2 \log \hat{\mathbb{P}}(\tau_1 \prec \tau_2)\Big] \tag{4}$$

Complementing the pairwise comparisons, binary safety label $\epsilon$ is also collected in constraint inference works (Reddy Chirra et al., 2024; Dai et al., 2024), enabling the model to explicitly distinguish safe and unsafe trajectories. $\epsilon = 1$ denotes safe trajectories satisfying the true constraint $C(\tau) \leq d$, otherwise $\epsilon = 0$. The full preference dataset is thus defined as $\mathcal{D} = \{(\tau_1, \tau_2, \mu_1, \mu_2, \epsilon_1, \epsilon_2)\}$.

Our goal is to infer the safety constraint (Eq. 2b) by learning $\hat{c}(s, a)$ from $\mathcal{D}$. To address the issues identified in Section 1, we propose a composite loss function for Preference-based Constrained Inference (PbCI):

$$\mathcal{L}_{PbCI} = \mathcal{L}_{pair} + \mathcal{L}_{safe}^{DZ} + \mathcal{L}_{SNR} \tag{5}$$

where $\mathcal{L}_{pair}$ is the pairwise preference loss in Eq. 4, $\mathcal{L}_{safe}^{DZ}$ is the dead-zone augmented safety loss (detailed in Section 3), and $\mathcal{L}_{SNR}$ is the SNR loss (introduced in Section 4). We will elaborate these in the following sections.

## 3. Constraint Inference with Dead Zone

### 3.1. Cost function learning

Given a preference dataset $\mathcal{D} = \{(\tau_1, \tau_2, \mu_1, \mu_2, \epsilon_1, \epsilon_2)\}$, we leverage the BT model to learn a cost function $\hat{c}(s, a)$. For two trajectories $\tau_1, \tau_2$ and pairwise preference labels $\mu_1, \mu_2$, the pairwise loss $\mathcal{L}_{pair}$ is formulated in Eq. 4 to capture the relative cost ranking.

Safety label $\epsilon$ can be used as a special case of pairwise preference label, because it relates to a pairwise comparison made between the given trajectory $\tau$ and a hypothetical threshold trajectory $\tau_{th}$, which has true cost $C(\tau_{th}) = d$ and estimated cost $\hat{C}(\tau_{th}) = 0$. Therefore, the probability that $\tau$ is considered safe can be expressed as the pairwise preference between $\tau_{th}$ and $\tau$:

$$\hat{\mathbb{P}}(\tau \text{ is safe}) = \hat{\mathbb{P}}(\tau \succ \tau_{th}) = \sigma(\hat{C}(\tau_{th}) - \hat{C}(\tau))$$
$$= \sigma(-\hat{C}(\tau)) \tag{6}$$

Similar to Eq. 4, the safety label $\epsilon$ and Eq. 6 can be applied to define a safety loss, facilitating the BT model to judge the safety of trajectory.

$$\mathcal{L}_{safe} = -\mathbb{E}_\mathcal{D}\Big[\epsilon \log \hat{\mathbb{P}}(\tau \text{ is safe}) + (1 - \epsilon) \log \hat{\mathbb{P}}(\tau \text{ is unsafe})\Big]$$
$$= -\mathbb{E}_\mathcal{D}\Big[\epsilon \log \sigma(-\hat{C}(\tau)) + (1 - \epsilon) \log \sigma(\hat{C}(\tau))\Big] \tag{7}$$

A well-trained BT model should perfectly rank the costs of all trajectories, and yield $\hat{C}(\tau) \leq 0$ for all safe trajectories with safety label $\epsilon = 1$, and $\hat{C}(\tau) > 0$ otherwise.

However, such perfect ranking and classification of trajectory costs may not guarantee that the learned constraint is aligned with the true underlying constraint. As stated in Section 1, BT model is designed for rank learning and it struggles with capturing absolute values or distribution.

As shown in Figure 1, the difference between the true heavy-tailed distribution and the relatively symmetric one, learned by the BT model, may lead to an underestimation of the cost expectation. Since the constraint is modeled in the expectation form $\mathbb{E}[C] \leq d$, this underestimation may further loosen the safety constraint and eventually fail to guide the policy learning for true safety requirements.

To address this constraint misalignment issue, we turn to use dead zone to mitigate the distribution difference.

The dead zone is a region around the threshold with upper bound $\delta > 0$. Instead of merely learning that unsafe trajectories with $\epsilon = 0$ should have $\hat{C} > 0$, we encourage the model to induce higher costs $\hat{C} > \delta$ for unsafe trajectories, pushing cost on the right side of the threshold to distribute further. With the above updates, the safety loss is refined as follows:

$$
\begin{aligned}
\mathcal{L}_{safe}^{DZ} &= -\mathbb{E}_{\mathcal{D}}\Big[\epsilon \log \hat{\mathbb{P}}(\tau \text{ is safe}) + (1-\epsilon) \log \hat{\mathbb{P}}(\tau \text{ is unsafe})\Big] \\
&= -\mathbb{E}_{\mathcal{D}}\Big[\epsilon \log \sigma(-\hat{C}(\tau)) + (1-\epsilon) \log \sigma(\hat{C}(\tau) - \delta)\Big]
\end{aligned}
\tag{8}
$$

The dead zone upper bound $\delta$ assigns higher costs to unsafe trajectories, producing a more heavy-tailed cost distribution. This mitigates the underestimation of expected cost for better constraint alignment.

### 3.2. Theoretical Analysis of Dead Zone

In this section, we present a theoretical analysis demonstrating how our dead-zone loss recovers the heavy-tailed cost distribution. This framework establishes a progression from micro-optimization to macro-statistics: Lemma 3.1 proves the gradient advantage of the dead-zone loss, which Theorem 3.2 propagates into multi-step dominance. Finally, Corollary 3.3 leverages this cost dominance to guarantee the recovery of the macroscopic heavy-tailed distribution.

**Lemma 3.1.** *For any unsafe trajectory $\tau$ with safety label $\epsilon = 0$ and estimated cost $\hat{C}(\tau)$, the gradient provided by the Dead Zone loss $\mathcal{L}_{safe}^{DZ}$ in Eq. 8 is strictly more negative than that provided by the original loss $\mathcal{L}_{safe}$ in Eq. 7, i.e., $\nabla_{\hat{C}(\tau)}\mathcal{L}_{safe}^{DZ} < \nabla_{\hat{C}(\tau)}\mathcal{L}_{safe} < 0$.*

*Proof.* See Appendix B.1. □

Building on the gradient difference in Lemma 3.1, we analyze the cumulative effect of multistep gradient descent over the training process of the cost model.

**Theorem 3.2.** *For any unsafe trajectory $\tau$, $\hat{C}_t^{DZ}(\tau)$ and $\hat{C}_t(\tau)$ are the cost learned by the dead zone loss $\mathcal{L}_{safe}^{DZ}$ and the original loss $\mathcal{L}_{safe}$ at training step $t$, respectively. Under gradient descent with the same learning rate $\eta$ and identical initialization $\hat{C}_0^{DZ}(\tau) = \hat{C}_0(\tau)$, for any $t > 0$, we have strict dominance $\hat{C}_t^{DZ}(\tau) > \hat{C}_t(\tau)$.*

*Proof.* We analyze the update dynamics using mathematical induction. Detailed proof is provided in Appendix B.2. □

Finally, we connect this instance-level cost shift to the overall distributional change induced by the dead zone.

**Corollary 3.3.** *The cost distribution learned by the dead zone loss $\mathcal{L}_{safe}^{DZ}$ has a strictly heavier right tail than that learned by the original loss $\mathcal{L}_{safe}$. Specifically, for any $z > 0$ (within the support), the tail probability satisfies $\mathbb{P}(\hat{C}^{DZ} \geq z) > \mathbb{P}(\hat{C} \geq z)$.*

*Proof.* See Appendix B.3. □

The above analysis shows that the dead zone upper bound $\delta$ encourages the learned cost distribution to have a heavier right tail, mitigating the underestimation of expected cost. This will facilitate better constraint alignment and policy optimization in downstream safe RL tasks.

## 4. Loss Design based on Signal-to-Noise Ratio

The dead zone mechanism from Section 3 is primarily designed to align the constraint with the true underlying constraint. However, there is evidence showing that a cost model with a flattened or poorly differentiated cost landscape may hinder effective policy gradient updates, leading to suboptimal performance or slow convergence(Razin et al., 2025). This section is focused on how to induce cost differentiation and send informative signals for efficient policy optimization.

Specifically, we encourage the cost model to search for better differentiation by adding a novel loss term based on Signal-to-Noise Ratio (SNR). The SNR is a well-known measure defined by the signal power relative to the noise power(Johnson, 2006). Here, we define the SNR using a batch of data $\{(\tau, \mu)\}$ from the dataset $\mathcal{D}$ as follows:

$$
\mathcal{L}_{SNR} = -\zeta \frac{Var(\hat{C}(\tau))}{\mathcal{H}(p(\mu))}
\tag{9}
$$

where $\zeta$ is a weighting coefficient. $Var(\hat{C}(\tau))$ is the variance of the estimated cost of the batch, which is the signal power because it reflects the degree of differentiation in the cost outputs. $\mathcal{H}(p(\mu)) = -\sum p(\mu) \ln p(\mu)$ is the entropy of the preference label distribution $p(\mu)$, which quantifies the uncertainty in the preference data and serves as the noise power, because larger entropy indicates greater disorder and less information provided by the labels.

Minimizing the SNR loss encourages the model to induce sufficient cost variance. It thus provides a more discriminative cost signal for policy optimization, with a benefit of policy exploration. We will elaborate this further in Appendix C.

Incorporating pairwise preference loss in Eq. 4, safety loss in Eq. 8 and SNR loss in Eq. 9, the overall loss function for preference-based constrained inference $\mathcal{L}_{PbCI}$ is defined in Eq. 5. The overall loss function is optimized to learn a

cost function $\hat{c}(s, a)$ that not only aligns well with the true underlying constraint but also provides informative signals for efficient policy optimization.

# 5. Two-stage Training Strategy

This section will propose a two-stage training strategy for the cost model, and then present the overall Preference-based Constrained Reinforcement Learning (PbCRL) algorithm and its theoretical convergence guarantee.

## 5.1. Constraint Learning Details

Most preference-based RL methods jointly optimize the cost model and the policy online (Reddy Chirra et al., 2024; Lee et al., 2021; Christiano et al., 2017). Inspired by reward model training in language model alignment (Ouyang et al., 2022), we instead propose a two-stage learning strategy. This approach leverages offline data for initial cost model training, followed by online finetuning during policy optimization, thereby lowering the burden of online labeling and enhancing constraint satisfaction.

**Offline Pre-training** Standard online approaches require annotators to continuously provide feedback throughout the entire policy training procedure, resulting in a significant labeling burden that is often expensive and time-consuming. To mitigate this, our first stage leverages an offline preference dataset $\mathcal{D}$ collected in advance. The cost model $c_\psi(s, a)$, parameterized by $\psi$, is trained by the loss in Eq. 5 with fixed dead zone for stable training.

$$\psi \leftarrow \psi - lr_\psi \nabla_\psi \mathcal{L}_{PbCI}(\psi) \tag{10}$$

This procedure leverages existing preference data without incurring online labeling. The offline pre-training phase aims to learn a general estimate of the cost function based on the available offline data.

**Online Finetuning** The pre-trained cost model is then used for policy optimization in Eq. 2. During this policy training phase, we select a comparatively small batch of generated trajectories for online labeling to finetune the cost model by Eq. 10.

Furthermore, the dead zone parameter fixed during pre-training may not optimally transfer to the online phase, due to the distribution shift between the offline dataset and the current policy's trajectories. To address this, we introduce an adaptive calibration mechanism that aligns the model's safety predictions with the empirical ground truth.

For a batch of trajectories with online labels $\mathcal{B} = \{(\tau, \mu, \epsilon)\}$, while the ground truth cost values are unavailable, the violation rate (proportion of unsafe trajectories) provides a surrogate signal for alignment. We aim to match the pre-

dicted violation rate $\hat{\mathbb{P}}_{vio} = \frac{1}{|\mathcal{B}|} \sum \mathbb{I}(c_\psi(\tau) > 0)$ with the empirical violation rate $\mathbb{P}_{vio} = \frac{1}{|\mathcal{B}|} \sum \mathbb{I}(\epsilon = 0)$. A discrepancy where $\hat{\mathbb{P}}_{vio} > \mathbb{P}_{vio}$ indicates an overestimation of risks, while the inverse suggests underestimation. We adaptively update the dead zone to minimize this mismatch:

$$\mathcal{L}_\delta = \|\hat{\mathbb{P}}_{vio} - \mathbb{P}_{vio}\|^2 \tag{11}$$

We perform gradient descent on $\mathcal{L}_\delta$ with respect to $\delta$ to update the dead zone. This process acts as a dynamic calibration, ensuring that the learned cost model maintains a consistent interpretation of safety with preference feedback as the policy evolves.

The two-stage learning phase leverages offline data for pre-training, reducing the need for costly online labeling. The adaptive finetuning of the cost model during policy training allows it to align the pre-trained cost function with the genuine safety requirements encountered in the training environment, thereby adaptively enhancing constraint satisfaction of the learned policy.

## 5.2. Preference-based Constrained Reinforcement Learning (PbCRL)

Based on these, we present the proposed algorithm, namely Preference-based Constrained Reinforcement Learning (PbCRL). PbCRL leverages an extended BT model to estimate safety constraint from preference. Policy optimization is then performed based on this learned constraint. In this work, we conduct policy optimization using the Lagrange multiplier method, a widely adopted approach in safe RL literature (Ray et al., 2019; Yang et al., 2021; Li et al., 2025). The safe RL problem in Eq. 2 is transformed into a Lagrangian loss, with cost model parameter $\psi$, policy parameter $\theta$ and Lagrange multiplier $\lambda$:

$$\mathcal{L}(\psi, \theta, \lambda) = -\Big[\mathcal{J}^R(\pi_\theta) - \lambda \mathcal{J}^{C_\psi}(\pi_\theta)\Big] \tag{12}$$

The overall algorithm of the proposed PbCRL is summarized in Algorithm 1.

## 5.3. Convergence Analysis of PbCRL

**Assumption 5.1.** The following conditions are adopted as standard assumptions in Safe RL literature (Borkar, 2008; Chow et al., 2018; Li et al., 2025):

1. **Continuous:** The objective $\mathcal{L}(\psi, \theta, \lambda)$ in Eq. 12 is continuously differentiable w.r.t. $\theta$, and its gradient $\nabla_\theta \mathcal{L}(\psi, \theta, \lambda)$ is Lipschitz continuous w.r.t. $\psi$, $\theta$ and $\lambda$.

2. **Step Sizes and Timescale Separation:** The learning rates $lr_\psi$, $lr_\theta$ and $lr_\lambda$ are positive, nonsummable, and square-summable. Additionally, they satisfy timescale

**Algorithm 1** Preference-based Constrained Reinforcement Learning (PbCRL)

---

**Require:** Preference dataset $\mathcal{D}$, finetune interval $K$,
 1: **Cost Model Pre-training:**
 2: **Initialize** cost model $c_\psi(s, a)$, dead zone $\delta$
 3: **for** each epoch **do**
 4:     Sample a batch of preference data from $\mathcal{D}$
 5:     Update cost model parameter $\psi$ by Eq. 10
 6: **end for**
 7:
 8: **Policy Optimization:**
 9: **Initialize** policy $\pi_\theta$, Lagrange multiplier $\lambda$
10: **for** $n = 0, 1, 2, \ldots, N$ **do**
11:     Collect trajectories using current policy $\pi_\theta$
12:     **if** $n \bmod K = 0$ **then**
13:         **Cost Model Finetuning:**
14:         Sample a small batch of online data $\mathcal{B}$, update $\mathcal{D}$
15:         Update dead zone $\delta$ by Eq. 11
16:         **for** each epoch **do**
17:             Sample a batch of preference data from $\mathcal{D}$
18:             Update cost model parameter $\psi$ by Eq. 10
19:         **end for**
20:     **end if**
21:     Update policy $\pi_\theta$ by $\theta \leftarrow \theta - lr_\theta \nabla_\theta \mathcal{L}(\psi, \theta, \lambda)$
22:     Update multiplier $\lambda$ by $\lambda \leftarrow \lambda + lr_\lambda \nabla_\lambda \mathcal{L}(\psi, \theta, \lambda)$
23: **end for**

---

separation conditions: $lr_\lambda(t) = o(lr_\theta(t)), lr_\theta(t) = o(lr_\psi(t))$.

3. **Stability of Fast Dynamics:** For any fixed $\theta$ and $\lambda$, the Ordinary Differential Equation (ODE) governing the cost model finetuning $\dot{\psi} = -\nabla_\psi \mathcal{L}_{PbCI}(\psi)$ possesses a set of locally asymptotically stable equilibria, and the iterates $\{\psi_k\}$ remain almost surely within the domain of attraction of this set.

With these assumptions, we present the convergence theorem of PbCRL as follows:

**Theorem 5.2.** *Under Assumptions 5.1, PbCRL with iterations $(\psi, \theta, \lambda)$ converges almost surely to a local optimal policy for the safe RL problem in Eq. 2.*

*Proof.* We utilize multi-time scale stochastic approximation to prove the convergence. Details are in Appendix B.4.  □

Theorem 5.2 shows that under standard assumptions, the proposed PbCRL algorithm converges to a local optimum of the safe RL problem in Eq. 2.

# 6. Experiments

In this section, we empirically evaluate the proposed PbCRL on multiple benchmarks to answer the following research questions:

**Q1:** Can PbCRL effectively learn safety constraints from preference data and achieve superior performance compared to state-of-the-art baselines?

**Q2:** Why does PbCRL work? How do the proposed techniques contribute to the overall performance of PbCRL?

**Q3:** Can PbCRL generalize to more complex scenarios?

**Benchmarks** We evaluate PbCRL on three distinct domains with varied safety challenges: (1) Robotic control in Safety Gymnasium (Ji et al., 2023); (2) Autonomous driving; (3) Language Model alignment. Details for each domain are provided in the respective subsections.

**Baselines and Metrics** We compare PbCRL against state-of-the-art baselines in Safe RL with constraint learned from preference. **RLSF** (Reddy Chirra et al., 2024) estimates a binary cost function using a BT model trained from segment-level human feedback. Safe RLHF (Dai et al., 2024), originally for language model alignment, is adapted for our continuous control tasks using its standard BT-based cost learning. We refer to this baseline as **PPO-BT** to highlight its reliance on the standard BT model. Additionally, we include **PPO-Lag** (Ray et al., 2019) trained with ground-truth costs as an oracle upper bound. A safe RL approach with well-aligned constraints should approach the performance of PPO-Lag. Performance is evaluated using average episode **return** and ground-truth **cost** over 5 random seeds. Implementation details are provided in Appendix D.

## 6.1. Robotic tasks in Safety Gymnasium

**Environment Setup** We evaluate PbCRL on robotic control tasks in Safety Gymnasium (Ji et al., 2023), a standard safe RL benchmark. Locomotion tasks (HalfCheetah, Walker2d, Humanoid) involve velocity constraints, while the more challenging navigation tasks (Goal, Push, Button) require reaching targets while avoiding collisions with obstacles. Cost thresholds are scaled with task difficulty following standard Safe RL protocols. Full details are in Appendix D.1.

**Overall Performance** To answer **Q1** regarding the effectiveness and superiority of our approach, we first present a holistic evaluation across six Safety Gymnasium tasks in Table 1. PbCRL demonstrates superior results by effectively balancing return and safety, outperforming other preference-based baselines. Crucially, the true cost of PbCRL converges closely to the safety threshold, mirroring the behavior of the Oracle (PPO-Lag). This empirically validates that the constraint learned by PbCRL functions as a reliable surrogate for the underlying safety constraint.

*Table 1.* Empirical results on Safety Gymnasium tasks. The best method using learned constraints is highlighted in **bold**.

| Tasks (Threshold) | Metrics | PPO-Lag (Oracle) | PbCRL (Ours) | RLSF | PPO-BT |
|---|---|---|---|---|---|
| HalfCheetah (5) | Return | $2619 \pm 124$ | $\mathbf{2367 \pm 138}$ | $2084 \pm 126$ | $2494 \pm 195$ |
| | Cost | $4.82 \pm 0.91$ | $4.66 \pm 1.03$ | $3.26 \pm 0.78$ | $5.58 \pm 2.14$ |
| Walker2d (5) | Return | $3057 \pm 142$ | $2659 \pm 133$ | $2415 \pm 120$ | $\mathbf{2807 \pm 156}$ |
| | Cost | $4.88 \pm 1.02$ | $4.62 \pm 0.97$ | $3.66 \pm 0.85$ | $\mathbf{4.91 \pm 1.11}$ |
| Humanoid (5) | Return | $5876 \pm 210$ | $\mathbf{5247 \pm 149}$ | $4816 \pm 180$ | $5948 \pm 225$ |
| | Cost | $5.03 \pm 1.93$ | $4.85 \pm 1.21$ | $4.05 \pm 1.12$ | $5.89 \pm 2.05$ |
| Goal (20) | Return | $23.50 \pm 1.14$ | $\mathbf{23.41 \pm 1.08}$ | $5.78 \pm 1.07$ | $24.26 \pm 0.99$ |
| | Cost | $20.70 \pm 1.91$ | $\mathbf{18.50 \pm 2.03}$ | $8.20 \pm 2.21$ | $26.97 \pm 2.62$ |
| Push (35) | Return | $4.77 \pm 0.67$ | $\mathbf{3.20 \pm 0.38}$ | $1.07 \pm 0.35$ | $4.09 \pm 0.57$ |
| | Cost | $36.17 \pm 5.00$ | $\mathbf{34.86 \pm 3.13}$ | $20.05 \pm 3.56$ | $41.85 \pm 5.65$ |
| Button (40) | Return | $8.96 \pm 1.06$ | $\mathbf{8.73 \pm 0.78}$ | $-2.01 \pm 0.37$ | $9.01 \pm 1.51$ |
| | Cost | $38.25 \pm 5.20$ | $\mathbf{37.51 \pm 5.65}$ | $4.26 \pm 3.33$ | $50.78 \pm 8.10$ |

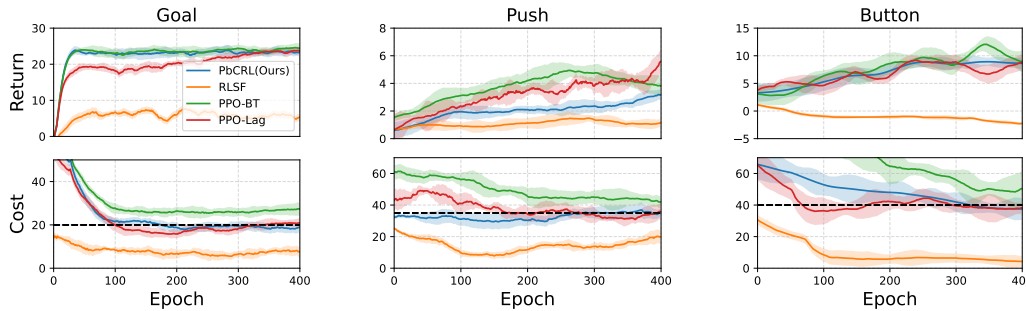

*Figure 2.* Average episode return and cost of four algorithms on robotic navigation tasks. Dashed lines are safety thresholds.

To provide deeper insight into how these results are achieved, Figure 2 illustrates the training dynamics on three representative navigation tasks. We observe that the true cost (Row 2) of all algorithms converges towards a steady state, indicating all policies satisfy their safety constraints, either learned (PbCRL, RLSF, PPO-BT) or ground truth (PPO-Lag).

PPO-Lag (red), the oracle with ground-truth costs, establishes a performance upper bound with high returns and satisfied constraints. In contrast, RLSF (orange) yields the lowest return and cost across all methods. This behavior is likely due to an overestimation issue in its cost function learned from segments, resulting in overly conservative constraints that sacrifice return performance for excessive safety (Reddy Chirra et al., 2024).

Comparing PbCRL (blue) with PPO-BT (green), PbCRL achieves high returns comparable to the Oracle PPO-Lag (red) while maintaining costs just below the safety threshold (black dashed line). This balance between return and safety, similar to PPO-Lag, demonstrates that the learned constraint successfully aligns with the genuine safety requirements. Conversely, PPO-BT, despite its high return, consistently violates constraints with costs exceeding the threshold. As analyzed in Section 3, this failure is attributed to the standard BT model's tendency to infer symmetric cost distributions rather than heavy-tailed ones, causing underestimation of the expected cost and overly aggressive policies.

**Cost Distribution Analysis**  Addressing **Q2** on why PbCRL works, we delve into the characteristics of the learned cost distributions to explain the performance disparity. As illustrated in Figure 1, an accurate representation of the underlying cost distribution, especially its heavy-tailed nature, is vital for learning a reliable constraint that aligns with the true constraint $\mathbb{E}[C] < d$. We quantify the fidelity of learned cost distributions using the 2-Wasserstein (W2) distance against the ground truth, a lower W2 distance implies higher similarity between two distributions.

*Table 2.* W2 and Bias between converged cost and threshold

| Tasks | PbCRL (Ours) | | RLSF | | PPO-BT | |
|---|---|---|---|---|---|---|
| | W2 | Bias | W2 | Bias | W2 | Bias |
| Goal | **16.8** | **1.5** | 68.1 | 11.8 | 42.7 | 6.9 |
| Push | **20.3** | **0.1** | 82.4 | 14.9 | 36.5 | 6.8 |
| Button | **44.3** | **2.5** | 165.5 | 35.7 | 78.8 | 10.78 |

Table 2 summarizes W2 distances and the bias between the converged true cost and the safety threshold. The results reveal a strong correlation: lower W2 distances consistently correspond to lower safety bias, indicating better constraint alignment. PbCRL achieves the lowest W2 across all tasks, confirming that the dead zone mechanism successfully encourages heavy-tailed characteristics. This distributional alignment directly explains the superior constraint satisfaction observed in Table 1. Further analysis is provided in Appendix F.2.

### 6.2. Generalization to Complex Scenarios

To address **Q3** regarding whether PbCRL can generalize to more complex and diverse domains, we extend our evaluation to two distinct settings: autonomous driving (highly

dynamic continuous control) and language model alignment (high-dimensional discrete action space).

**Autonomous Driving** We utilize an autonomous driving simulator (Erdem et al., 2020) featuring two highway tasks: blocked road and lane change. These tasks require the agent to navigate dynamic traffic while avoiding collisions in highly interactive and uncertain environments. As shown in Table 3, PbCRL mirrors the Oracle's (PPO-Lag) performance, achieving high returns while strictly maintaining costs within the safety threshold. Conversely, PPO-BT consistently exceeds the safety threshold, while RLSF falls into conservatism with lower returns.

*Table 3.* Results on driving scenarios, R: Return, C: Cost

| Task(Threshold) | M | PPO-Lag(Oracle) | PbCRL(Ours) | RLSF | PPO-BT |
|---|---|---|---|---|---|
| Block (0.1) | R | $39.1 \pm 0.1$ | $\mathbf{36.4 \pm 0.3}$ | $31.8 \pm 0.3$ | $38.5 \pm 0.6$ |
| | C | $0.05 \pm 0.01$ | $\mathbf{0.07 \pm 0.01}$ | $0.05 \pm 0.01$ | $0.12 \pm 0.02$ |
| Lane (0.1) | R | $51.6 \pm 0.9$ | $\mathbf{51.4 \pm 0.5}$ | $44.9 \pm 0.8$ | $51.5 \pm 0.8$ |
| | C | $0.01 \pm 0.01$ | $\mathbf{0.08 \pm 0.01}$ | $0.06 \pm 0.01$ | $0.14 \pm 0.02$ |

**Language Model Alignment** While our primary focus remains on safety-critical continuous control, we conduct preliminary experiments in language model alignment to evaluate the generality of PbCRL in high-dimensional, discrete action spaces. Llama-3.2-1B is employed as base model and trained on Safe RLHF dataset (Dai et al., 2024). We utilize Gemini 2.5 Flash as an impartial judge to compute win rates, using consistent reward and cost models across all baselines. Results in Table 4 confirm PbCRL's potential to generalize to higher-dimensional domains.

*Table 4.* Results on Language Model Alignment

| Metric | PPO | RLSF | Safe RLHF | PbCRL (Ours) |
|---|---|---|---|---|
| Win Rate (Helpful) ↑ | 72.5% | 60.4% | 79.4% | **80.7%** |
| Win Rate (Harmless) ↑ | 60.7% | 54.2% | 76.5% | **82.1%** |
| Reward ↑ | 2.78 | 1.31 | 4.05 | **4.22** |
| Cost ↓ | $-0.57$ | 1.10 | $-2.97$ | $\mathbf{-3.03}$ |

## 6.3. Ablation Study

**Dead Zone Parameter** We investigate the sensitivity of the dead zone parameter $\delta$ during the cost model pre-training phase, with results summarized in Table 5.

*Table 5.* Ablation study on dead zone parameter $\delta$

| Tasks | M | $\delta = 0$ | $\delta = 0.1$ | $\delta = 1$ | $\delta = 2$ |
|---|---|---|---|---|---|
| | W2 | 38.9 | 33.5 | **16.8** | 29.1 |
| Goal (20) | R | $24.55 \pm 1.01$ | $24.79 \pm 1.18$ | $\mathbf{23.41 \pm 1.08}$ | $22.51 \pm 1.05$ |
| | C | $28.05 \pm 2.55$ | $27.46 \pm 1.89$ | $\mathbf{18.50 \pm 2.03}$ | $16.49 \pm 2.18$ |
| | W2 | 36.1 | 35.8 | **20.3** | 23.6 |
| Push (35) | R | $4.15 \pm 0.55$ | $3.81 \pm 0.21$ | $\mathbf{3.20 \pm 0.38}$ | $2.96 \pm 0.35$ |
| | C | $42.50 \pm 5.50$ | $40.26 \pm 2.15$ | $\mathbf{34.86 \pm 3.13}$ | $30.91 \pm 1.78$ |

Setting $\delta = 0$ reverts to the standard BT model. As discussed in Section 3, this induces a symmetric distribution

that underestimates tail risks and yields high W2 distances. A small offset ($\delta = 0.1$) marginally improves results, but remains insufficient. Conversely, a large dead zone ($\delta = 2$) over-corrects the distribution, leading to over-conservatism and distributional distortion (increased W2 distances). A moderate $\delta = 1$ achieves the lowest W2 and best reward-safety trade-off by accurately recovering the heavy-tailed cost distribution.

**Safety Threshold Offsets** A potential heuristic to address the risk underestimation of standard BT is manually lowering its safety threshold $d$. To evaluate this, we train PPO-BT on the *Goal* task (true threshold $d = 20$) with various stricter training targets $d \in \{20, 18, 15, 10\}$.

*Table 6.* Ablation study on varying safety threshold offsets

| M | PbCRL | PPO-BT (varying $d$) | | | |
|---|---|---|---|---|---|
| | $d = 20$ | $d = 20$ | $d = 18$ | $d = 15$ | $d = 10$ |
| R | $\mathbf{23.41 \pm 1.08}$ | $24.26 \pm 0.99$ | $23.86 \pm 1.21$ | $23.15 \pm 1.05$ | $18.27 \pm 1.13$ |
| C | $\mathbf{18.50 \pm 2.03}$ | $26.97 \pm 2.62$ | $24.08 \pm 2.87$ | $20.82 \pm 2.05$ | $14.02 \pm 1.95$ |

As summarized in Table 6, while manually lowering the threshold indeed reduces PPO-BT's actual cost, it yields a suboptimal reward-safety trade-off compared to PbCRL. Crucially, this heuristic may not be a viable solution in real-world deployment. Under preference-based settings, the true safety limit is implicitly embedded within the collected preference labels, rather than explicitly provided. Mathematically, simply shifting the threshold acts as a mere linear translation of a structurally biased, symmetric distribution. In contrast, PbCRL fundamentally resolves this mismatch by dynamically shaping the distribution to recover the heavy tail, achieving constraint satisfaction without sensitive, task-specific threshold tuning.

**SNR-based Loss** The SNR-based loss (Section 4) in Section 4 is designed to induce sufficient cost variance for efficient policy optimization. We conduct a sensitivity analysis on the SNR loss weight $\zeta$ in Goal task, summarizing mid-training and final performance in Table 7.

*Table 7.* Ablation study on SNR loss weight $\zeta$

| Metrics | $\zeta = 10^{-2}$ | $\zeta = 10^{-3}$ (Ours) | $\zeta = 10^{-5}$ | $\zeta = 0$ |
|---|---|---|---|---|
| Return (Mid) | $3.8 \pm 2.2$ | $\mathbf{22.9 \pm 1.5}$ | $22.5 \pm 1.3$ | $22.1 \pm 1.1$ |
| Cost (Mid) | $17.9 \pm 8.1$ | $\mathbf{18.4 \pm 2.3}$ | $24.5 \pm 2.5$ | $24.8 \pm 2.8$ |
| Return (End) | $6.5 \pm 1.8$ | $\mathbf{23.4 \pm 1.1}$ | $20.6 \pm 1.6$ | $20.7 \pm 1.4$ |
| Cost (End) | $21.9 \pm 5.9$ | $\mathbf{18.5 \pm 2.0}$ | $19.2 \pm 2.2$ | $18.4 \pm 2.6$ |

Results show that an excessive SNR weight ($10^{-2}$) destabilizes the learning process, evidenced by high variance in both return and cost. Conversely, removing (0) or underweighting ($10^{-5}$) the SNR term leads to a flattened cost landscape, resulting in slow safety convergence (higher costs at mid-training). A balanced $\zeta$ ($10^{-3}$) induces sufficient

cost differentiation, accelerating convergence with lower cost and higher return both at mid-training and final. This empirically validates that the SNR loss provides a more informative signal, facilitating more efficient policy learning.

**Two-stage Training Strategy** We compare PbCRL against PbCRL-Offline (cost model trained at offline stage only) in Goal and Push tasks, summarized in Table 8. While both methods leverage offline dataset and reduce online labeling burden, the full PbCRL achieves converged costs closer to the safety threshold. This confirms that the online finetuning stage enhances the alignment of the learned constraint, thereby improving constraint satisfaction.

*Table 8.* Ablation study on two-stage training strategy

| Tasks | Metrics | PbCRL-Offline | PbCRL |
|---|---|---|---|
| Goal (20) | Return | $24.79 \pm 1.75$ | $\mathbf{23.41 \pm 1.08}$ |
| | Cost | $22.70 \pm 0.84$ | $\mathbf{18.50 \pm 2.03}$ |
| Push (35) | Return | $3.77 \pm 0.67$ | $\mathbf{3.20 \pm 0.38}$ |
| | Cost | $37.48 \pm 5.00$ | $\mathbf{34.86 \pm 3.13}$ |

Further experimental results and analyses are provided in Appendix F.

## 7. Conclusion

In this paper, we addressed the significant challenge of learning complex, implicit safety constraints in real-world Safe Reinforcement Learning applications. We proposed Preference-based Constrained Reinforcement Learning (PbCRL), a novel framework that learns safety constraints from human preference over trajectories. Our technical contributions include extending the Bradley-Terry model with dead zone to better align the learned cost distribution with the true underlying heavy-tailed costs and mitigate constraint underestimation. We introduced a novel SNR-based loss to promote cost variation, providing a more informative signal for efficient policy optimization. Additionally, our two-stage training strategy effectively reduces the reliance on costly online labeling by leveraging offline data, while adaptive online finetuning enhances constraint satisfaction in the deployment environment. Experimental results demonstrated that our method achieved superior alignment with genuine safety requirements and outperformed state-of-the-art baselines. Despite these advances, our work has certain limitations. For instance, the current framework assumes that the human feedback is free from noise or inconsistencies. Future work could explore enhancing the robustness of PbCRL against noisy human judgments or adversarial attacks.

## Impact Statement

This paper presents work whose goal is to advance the field of machine learning. There are many potential societal consequences of our work, none of which we feel must be specifically highlighted here.

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

# A. Related Work

This section provides a comprehensive overview of related literature in Safe Reinforcement Learning, Constraint Inference, and Preference-based Learning, highlighting the position and contributions of our proposed PbCRL framework.

**Safe Reinforcement Learning (Safe RL)**    Safe RL addresses the problem of learning optimal policies under safety constraints, typically formulated as Constrained Markov Decision Processes (CMDPs) (Altman, 2021). Existing methods can be broadly categorized into primal-dual methods (Ray et al., 2019; Ding et al., 2020) and projection-based methods (Achiam et al., 2017; Yang et al., 2020). Primal-dual approaches, such as PPO-Lagrangian (Ray et al., 2019), employ Lagrangian multipliers to dynamically balance the reward maximization and cost minimization objectives. Projection-based methods, such as CPO (Achiam et al., 2017) and PCPO (Yang et al., 2020), project the policy gradient onto a feasible set to ensure constraint satisfaction at each update step.

While these methods have achieved significant success, they predominantly rely on **explicitly defined cost functions**. However, in many real-world applications like autonomous driving and robotics, defining such explicit constraints is often subjective, implicit, or prohibitively expensive (Cosner et al., 2022; Reddy Chirra et al., 2024; Xu & Liu, 2024). Our work complements these foundational Safe RL algorithms by focusing on the inference of latent safety constraints from human feedback, rather than assuming they are given.

**Constraint Inference/Learning**    To bypass manual constraint specification with dedicated system engineering, a growing body of work focuses on learning safety constraints from data. Inverse Reinforcement Learning (IRL) techniques (Ng et al., 2000) have been adapted to infer constraints that rationalize expert demonstrations, often referred to as Inverse Constrained Reinforcement Learning (ICRL) (Scobee & Sastry, 2020; Malik et al., 2021; Xu & Liu, 2023; 2024). Early works utilized Maximum Entropy IRL frameworks to infer constraints that explain expert behaviors (Scobee & Sastry, 2020). Recent advancements include extending ICRL to high-dimensional control tasks (Malik et al., 2021), incorporating uncertainty estimation (Xu & Liu, 2023), and leveraging robustness principles (Xu & Liu, 2024). Other approaches integrate control theoretic guarantees, such as Control Barrier Functions (CBF) (Robey et al., 2020) or Hamilton-Jacobi reachability analysis (Yu et al., 2022), to learn safe sets from demonstrations. Despite their promise, these methods fundamentally depend on **fine-grained expert demonstrations**, which can be costly or impossible to collect in safety-critical scenarios where experts may not be available or perfect (Arora & Doshi, 2021).

Alternatively, constraints can be learned from feedback signals. However, prior works often impose restrictive assumptions, such as limiting constraints to specific function classes (e.g., linear or Gaussian Process-based) (Sui et al., 2015; Amani et al., 2021) or requiring dense state-level feedback (Bennett et al., 2023). These requirements limit scalability in complex environments. In contrast, our work leverages trajectory-level preferences, avoiding restrictive assumptions on the constraint form while being more sample-effective than expert demonstrations or dense labeling.

**Preference-based Reinforcement Learning (PbRL)**    PbRL replaces the scalar reward signal with human preferences over trajectory segments, enabling policy learning when reward engineering is difficult (Christiano et al., 2017; Lee et al., 2021). The standard paradigm involves learning a reward model, typically parameterized by a Bradley-Terry (BT) model (Bradley & Terry, 1952), from pairwise comparisons to guide policy optimization (Ouyang et al., 2022). While PbRL has been extensively studied for reward maximization, its application to constraint learning is relatively nascent.

Recent efforts have begun to apply BT-based models to infer safety constraints. For instance, Reddy Chirra et al. (2024) utilize BT models to learn binary cost functions from human preferences with various granularity, trajectory level in simple tasks or state level in complex tasks. Within the domain of language model alignment, works such as Dai et al. (2024) also leverage the BT model to constrain potentially harmful behaviors in model-generated text responses. However, these methods inherit the limitations of standard ranking models: they focus on relative ordering rather than absolute value estimation. As identified in our work, standard BT models tend to yield symmetric utility distributions, failing to capture the **heavy-tailed nature** of safety costs caused by cascading failures in continuous control tasks (Yang et al., 2021; Li et al., 2025). This mismatch leads to risk underestimation. Our PbCRL framework addresses this critical gap by introducing a dead-zone mechanism to enforce heavy-tailed cost distributions, ensuring accurate safety alignment.

# B. Proofs

## B.1. Proof of Lemma 3.1

*Proof.* Considering an unsafe trajectory $\tau$ with safety label $\epsilon = 0$ and estimated cost $\hat{C}(\tau)$, the gradient of the originally loss $\mathcal{L}_{safe}$ in Eq. 7 with respect to $\hat{C}(\tau)$ is given by:

$$\nabla_{\hat{C}(\tau)}\mathcal{L}_{safe} = \sigma(\hat{C}(\tau)) - 1 = -\sigma(-\hat{C}(\tau)) \tag{13}$$

The gradient of the Dead Zone loss $\mathcal{L}_{safe}^{DZ}$ with respect to $\hat{C}(\tau)$ is given by:

$$\nabla_{\hat{C}(\tau)}\mathcal{L}_{safe}^{DZ} = \sigma(\hat{C}(\tau) - \delta) - 1 = -\sigma(-(\hat{C}(\tau) - \delta)) \tag{14}$$

Since the sigmoid function is monotonically increasing and $\delta > 0$, we have:

$$-\sigma(-(\hat{C}(\tau) - \delta_{+})) < -\sigma(-\hat{C}(\tau)) < 0 \tag{15}$$

Therefore, $\nabla_{\hat{C}(\tau)}\mathcal{L}_{safe}^{DZ} < \nabla_{\hat{C}(\tau)}\mathcal{L}_{safe} < 0$ holds. $\qquad\square$

## B.2. Proof of Theorem 3.2

*Proof.* We analyze the update dynamics using mathematical induction. First define the gradient update operators for the two losses as:

$$F(\hat{C}) = \hat{C} - \eta\nabla_{\hat{C}}\mathcal{L}_{safe} = \hat{C} + \eta(1 - \sigma(\hat{C})) \tag{16}$$

$$F^{DZ}(\hat{C}) = \hat{C} - \eta\nabla_{\hat{C}}\mathcal{L}_{safe}^{DZ} = \hat{C} + \eta(1 - \sigma(\hat{C} - \delta)) \tag{17}$$

The cost updates at step $t$ can be expressed as:

$$\hat{C}_t(\tau) = F(\hat{C}_{t-1}(\tau)) \tag{18}$$

$$\hat{C}_t^{DZ}(\tau) = F^{DZ}(\hat{C}_{t-1}^{DZ}(\tau)) \tag{19}$$

We proceed with two properties to establish the induction.

**Operator dominance.** From Lemma 3.1, for any $\hat{C}$ and $\eta > 0$ we have

$$F(\hat{C}) < F^{DZ}(\hat{C}). \tag{20}$$

**Monotonicity.** Examining the derivative of $F^{DZ}(\hat{C})$ with respect to $\hat{C}$:

$$\frac{dF^{DZ}(\hat{C})}{d\hat{C}} = 1 - \eta\,\sigma(\hat{C} - \delta)\big(1 - \sigma(\hat{C} - \delta)\big). \tag{21}$$

Since $\sigma(x)(1 - \sigma(x)) \leq 0.25$ for all $x$, the condition $\eta < 4$ (which is satisfied by any standard learning rate configuration in practice) implies $\frac{dF^{DZ}(\hat{C})}{d\hat{C}} > 0$, i.e. $F^{DZ}(\hat{C})$ is strictly increasing.

**Induction step.** For the base case $t = 1$, from the identical initialization $\hat{C}_0^{DZ}(\tau) = \hat{C}_0(\tau)$ and operator dominance, we have:

$$\hat{C}_1^{DZ}(\tau) = F^{DZ}(\hat{C}_0^{DZ}(\tau)) > F(\hat{C}_0(\tau)) = \hat{C}_1(\tau). \tag{22}$$

Now assume $\hat{C}_k^{DZ}(\tau) > \hat{C}_k(\tau)$ holds for some $t = k$. From the monotonicity of $F^{DZ}$, we have:

$$F^{DZ}(\hat{C}_k^{DZ}(\tau)) > F^{DZ}(\hat{C}_k(\tau)). \tag{23}$$

From the operator dominance, we also have:

$$F^{DZ}(\hat{C}_k(\tau)) > F(\hat{C}_k(\tau)). \tag{24}$$

Combining the above two inequalities, we get:

$$\hat{C}_{k+1}^{DZ}(\tau) = F^{DZ}(\hat{C}_k^{DZ}(\tau)) > F(\hat{C}_k(\tau)) = \hat{C}_{k+1}(\tau). \tag{25}$$

Thus, by mathematical induction, the inequality $\hat{C}_t^{DZ}(\tau) > \hat{C}_t(\tau)$ holds for all $t > 0$. $\qquad\square$

**B.3. Proof of Corollary 3.3**

*Proof.* From Theorem 3.2, we have established that for every unsafe trajectory $\tau$, the learned cost values satisfy the strict inequality $\hat{C}^{DZ}(\tau) > \hat{C}(\tau)$. In probability theory, this pointwise strict dominance implies Strict First-Order Stochastic Dominance. By definition, if random variable $X$ dominates $Y$ in the first order, then for any non-decreasing function $u$, $\mathbb{E}[u(X)] > \mathbb{E}[u(Y)]$. Choosing the indicator function $u(c) = \mathbb{I}(c \geq z)$ (which represents the tail probability), we directly obtain:

$$\mathbb{P}(\hat{C}^{DZ} \geq z) > \mathbb{P}(\hat{C} \geq z)$$

This confirms that the dead zone loss assigns strictly higher probability mass to the tail region of the cost distribution. $\quad\square$

**B.4. Proof of Theorem 5.2**

*Proof.* We consider a three-time scale stochastic approximation framework:

$$\psi_{t+1} = \psi_t - lr_\psi(t)\nabla_\psi \mathcal{L}_{PbCI}(\psi_t)$$
$$\theta_{t+1} = \theta_t - lr_\theta(t)\nabla_\theta \mathcal{L}(\psi_t, \theta_t, \lambda_t)$$
$$\lambda_{t+1} = \lambda_t + lr_\lambda(t)\nabla_\lambda \mathcal{L}(\psi_t, \theta_t, \lambda_t) \tag{26}$$

where the cost model parameter $\psi$ is updated at the fastest timescale, followed by the policy $\theta$, and the Lagrange multiplier $\lambda$ at the slowest timescale. We analyze the convergence of each component sequentially.

**Convergence of cost model $\psi$.** From Assumption 5.1, the cost model update in Eq. 10 occurs at the fastest timescale. Thus, $\theta$ and $\lambda$ can be treated as quasi-static from the perspective of $\psi$. The update of $\psi$ tracks the ODE:

$$\dot{\psi} = -\nabla_\psi \mathcal{L}_{PbCI}(\psi) \tag{27}$$

In the online finetuning phase, the cost model parameters are initialized close to the previous solution. Under the timescale separation in Assumption 5.1, for fixed $\theta$ and $\lambda$, $\psi$ converges to the local attractor $\psi^*(\theta, \lambda)$ almost surely.

**Convergence of policy $\theta$.** The policy update occurs at the intermediate timescale. According to the timescale separation in Assumption 5.1, $\lambda$ can be considered as an arbitrary constant. To establish the convergence of the policy parameters, we invoke a local version of the two-time-scale convergence theorem (Adapted from (Borkar, 2008)):

**Theorem B.1.** *For a two timescale coupled iterations(Borkar, 2008):*

$$\psi_{t+1} = \psi_t + lr_\psi(t)(h(\psi_t, \theta_t) + m_t)$$
$$\theta_{t+1} = \theta_t + lr_\theta(t)(g(\psi_t, \theta_t) + n_t) \tag{28}$$

*for $t \geq 0$ satisfying:*

- $h(\psi, \theta)$ *and* $g(\psi, \theta)$ *are Lipschitz*

- *Noise sequences $m_t$, $n_t$ are bounded martingale differences*

- $lr_\psi(t), lr_\theta(t)$ *satisfy the step sizes timescale separation in Assumption 5.1*

*If for any fix $\theta$, ODE $\dot{\psi} = h(\psi, \theta)$ converges to a local attractor $\psi^*(\theta)$, then the iteration $\theta_k$ converges almost surely to the set of stationary points of the ODE:*

$$\dot{\theta} = g(\psi^*(\theta), \theta) \tag{29}$$

Viewed from the timescale of $\theta$, the cost model has equilibrated to $\psi^*(\theta)$. The policy update direction $g$ corresponds to the gradient of the Lagrangian $\mathcal{L}(\psi, \theta, \lambda)$. Thus, the asymptotic behavior of $\theta$ is governed by the ODE:

$$\dot{\theta} = -\nabla_\theta \mathcal{L}(\psi^*(\theta), \theta, \lambda) = -\nabla_\theta \mathcal{L}(\theta, \lambda) \tag{30}$$

This ODE describes a gradient descent flow on the Lagrangian loss. According to Theorem B.1, the sequence $\theta_k$ converges to the set of stationary points $\mathcal{Z} = \{\theta \mid \nabla_\theta \mathcal{L}(\theta, \lambda) = 0\}$. Since stochastic gradient descent algorithms avoid strict saddle points almost surely, we conclude that $\theta_k$ converges to a local minimizer $\theta^*(\lambda)$ of $\mathcal{L}(\theta, \lambda)$ almost surely.

**Convergence of multiplier** $\lambda$. Regarding the update of the Lagrange multiplier $\lambda$, as it follows the standard convergence analysis for Primal-Dual Safe RL algorithms (Borkar, 2008; Chow et al., 2018; Altman, 2021; Li et al., 2025), we provide a concise summary here.

Analogous to the analysis for $\theta$, the update of $\lambda$ can be viewed as a stochastic approximation process tracking the dual ascent trajectory. Established results in constrained MDP theory guarantee that, given the convergence of the faster-scale iterates ($\psi_k \to \psi^*$ and $\theta_k \to \theta^*$), the multiplier $\lambda_k$ converges almost surely to a stationary point.

Consequently, the full coupled system $(\psi_k, \theta_k, \lambda_k)$ converges to $(\psi^*, \theta^*, \lambda^*)$, a local saddle point of the Lagrangian $\mathcal{L}(\psi, \theta, \lambda)$. Integrating the saddle point properties of the Lagrangian, we conclude that $\pi_{\theta*}$ is a local optimal policy for the safe RL problem in Eq. 2. $\qquad\square$

## C. Further Analysis of SNR-based Loss

Recent research in Reinforcement Learning from Human Feedback (RLHF) (Razin et al., 2025) highlights that relying solely on prediction accuracy for reward model evaluation is insufficient. Specifically, a reward model with low variance tends to yield a flattened reward landscape, which hinders effective policy gradient updates. Motivated by this observation, the SNR-based loss proposed in Section 4 is designed to induce sufficient cost variance for efficient policy optimization. We demonstrate that maximizing the variance of the learned cost function (the numerator of the SNR) effectively raises the upper bound of the policy gradient norm in the downstream Safe RL phase, thereby facilitating efficient exploration.

Consider the gradient of the cost objective $\mathcal{J}^{C_\psi}(\pi_\theta)$ in Eq. (12) with respect to the policy parameters $\theta$:

$$\nabla_\theta \mathcal{J}^{C_\psi}(\pi_\theta) = \mathbb{E}_{\tau \sim \pi_\theta} \left[ C_\psi(\tau) \nabla_\theta \log \pi_\theta(\tau) \right] \tag{31}$$

Recall that the score function has a zero mean:

$$\mathbb{E}_{\tau \sim \pi_\theta} \left[ \nabla_\theta \log \pi_\theta(\tau) \right] = 0 \tag{32}$$

Let $\mu_C = \mathbb{E}\left[ C_\psi(\tau) \right]$. We can rewrite Eq. (31) using the definition of covariance:

$$\begin{aligned}
\nabla_\theta \mathcal{J}^{C_\psi}(\pi_\theta) =& \mathbb{E}_{\tau \sim \pi_\theta} \left[ C_\psi(\tau) \nabla_\theta \log \pi_\theta(\tau) \right] - \mu_C \mathbb{E}_{\tau \sim \pi_\theta} \left[ \nabla_\theta \log \pi_\theta(\tau) \right] \\
=& \text{Cov}(C_\psi(\tau), \nabla_\theta \log \pi_\theta(\tau))
\end{aligned} \tag{33}$$

Applying the Cauchy-Schwarz inequality, we derive the following upper bound on the gradient norm:

$$\|\nabla_\theta \mathcal{J}^{C_\psi}(\pi_\theta)\|^2 = \|\text{Cov}(C_\psi(\tau), \nabla_\theta \log \pi_\theta(\tau))\|^2 \leq \text{Var}(C_\psi(\tau)) \cdot \text{Var}(\nabla_\theta \log \pi_\theta(\tau)) \propto \text{Var}(C_\psi(\tau)) \tag{34}$$

The inequality in Eq. 34 indicates that maximizing the variance of the learned cost function through the SNR-based loss directly leads to an increased upper bound on the policy gradient norm. This ensures that the policy receives a more informative and diverse cost gradient signal during optimization, facilitating efficient exploration of the policy space and improving convergence in policy learning.

## D. Implementation Details

### D.1. Simulation Environments

**Safety Gymnasium Environments** Safety Gymnasium (Ji et al., 2023) is a suite of diverse simulated environments for safe reinforcement learning. We focus on two types of tasks within Safety Gymnasium: locomotion tasks and navigation tasks.

Locomotion tasks include three agents: HalfCheetah, Walker2d and Humanoid, shown in Figure 3. These tasks require the agent to move forward as fast as possible while satisfying velocity constraints.

The observation space includes position and velocity information of the agent's joints. The dimensionality of the observation space is 18 in HalfCheetah, 12 in Walker2d and 350 in Humanoid. The action space is continuous, representing the torques applied to the agent's joints. The reward function is defined by the forward velocity of the agent, with an additional penalty for taking too large actions. Safety violations are quantified by a cost of +1 for each time step the agent's velocity exceeds a predefined threshold, otherwise 0.

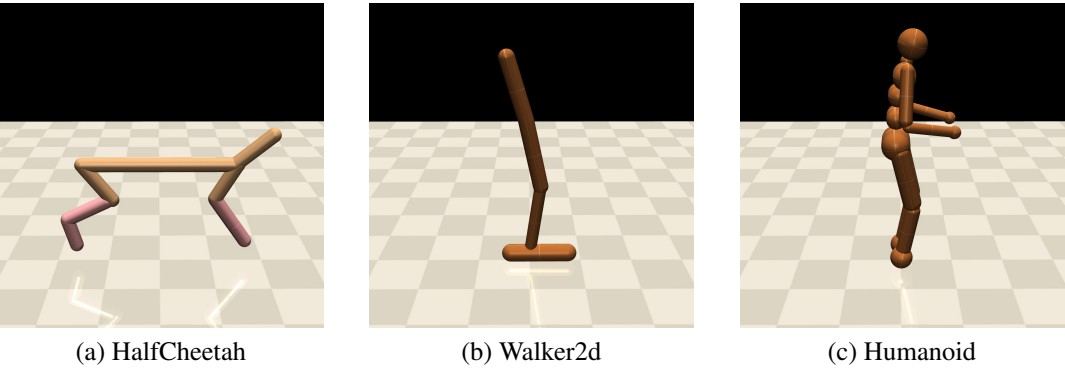

(a) HalfCheetah            (b) Walker2d            (c) Humanoid

*Figure 3.* Locomotion Environments in Safety Gymnasium

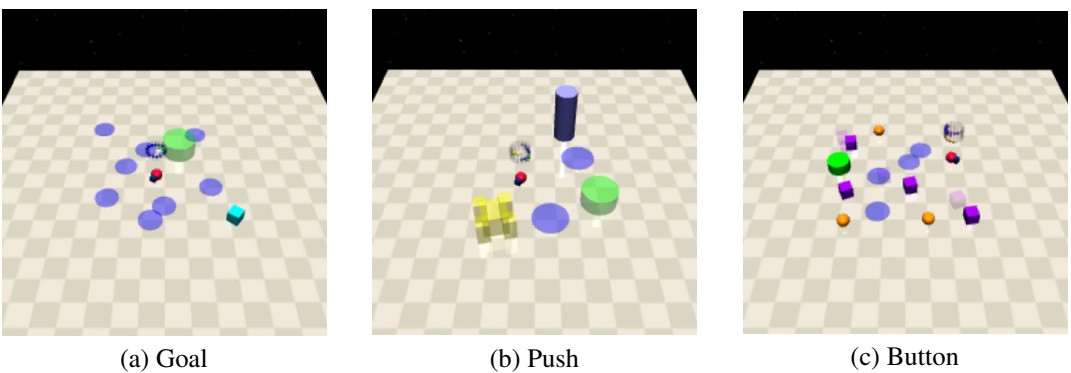

(a) Goal            (b) Push            (c) Button

*Figure 4.* Navigation Environments in Safety Gymnasium

Navigation tasks include three environments: Goal, Push, and Button. RL policies are trained to navigate a Point agent to a specified target while avoiding collisions with static and moving obstacles. Goal task in Figure 4 (a) requires the agent (red) to navigate a target location (green cylinder) while avoiding collisions with hazards (blue circle) and vases (cyan-blue cube). Push task in Figure 4 (b) involves pushing an object (yellow box) to a target location (green cylinder) while avoiding collisions with hazards (blue circle) and pillars (blue cylinder). Button task in Figure 4 (c) requires pressing a correct button (green cylinder) while avoiding collisions with hazards (blue circle), moving gremlins (purple cube) and other buttons (orange sphere).

The challenges of the three tasks increase in the order of Goal, Push, and Button. The Goal task is the simplest, where the agent only needs to reach a target location while avoiding static obstacles. The Push task adds an additional object that needs to be pushed to the target location, making it more complex. The Button task is the most challenging environment, as it involves both static and dynamic obstacles.

The observation space includes agent sensory information (accelerometer, velocimeter, gyro and magnetometer) and lidar data of objects in the environment. The dimensionality of the observation space is 60 in Goal, 76 in Push and Button. The action space is 2-dimensional, representing the agent's rotation and forward/backward movement. The reward function is defined by the distance reduction to the target at each step, with an additional positive reward upon reaching the target. Safety violations are quantified by a cost of +1 for each collision with obstacles, otherwise 0. When the agent manages to accomplish the task, the environment will regenerate a new task location randomly. This setup encourages the agent to complete the task as quickly as possible while avoiding collisions in an episode of 1000 steps. Environment will be reset after each episode.

Cost thresholds are assigned task-specifically that scale with difficulty, which is a standard protocol in Safe RL (Gu et al., 2024a) to ensure informative evaluation. Uniform thresholds are often uninformative: loose limits render simple tasks vacuous, while tight limits make complex tasks infeasible. Experiments with a uniform threshold is conducted in Appendix F.3 to confirm this.

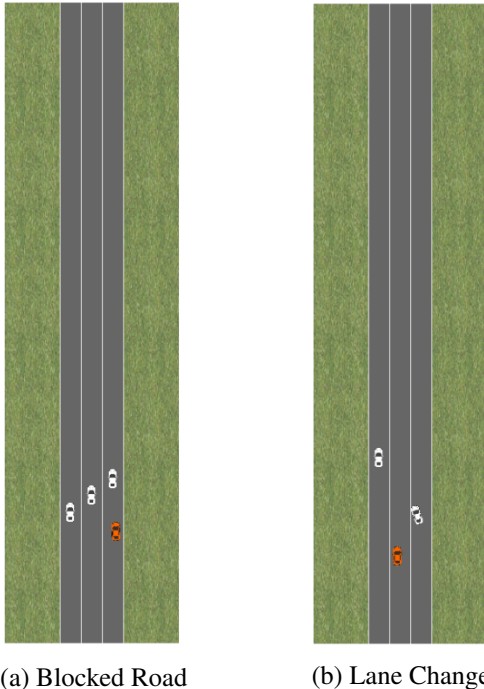

(a) Blocked Road          (b) Lane Change

*Figure 5.* Autonomous Driving Environments

We conducted all experiments on Ubuntu 20.04 with 24-core CPU, NVIDIA RTX 4090, and 128GB RAM. PbCRL requires approximately 5 hours for each individual run (1.5 hours for offline cost pre-training and 3.5 hours for online fine-tuning and policy training), whereas other fully online baselines require approximately 6 hours.

**Autonomous Driving Environment**    We utilize an autonomous driving simulator in (Erdem et al., 2020; Reddy Chirra et al., 2024). The environment simulates two highway driving scenarios: blocked road and lane change, as shown in Figure 5. The observation space consists of the ego vehicle's state (position, velocity, acceleration) and the surrounding vehicles' states (relative position, velocity, acceleration). The action space is the steering angle and acceleration of the ego vehicle. The reward function is defined based on the distance reduction to the destination at each step. The agent receives a positive cost for off-road driving, exceeding speed limits or getting too close to other vehicles.

**Language Model Alignment**    We include preliminary experiments on language model alignment to evaluate the generality of PbCRL in high-dimensional, discrete action spaces. We employ Llama-3.2-1B [1] as the base model and train it on the Safe RLHF dataset (Dai et al., 2024), which contains human preferences over model-generated text responses. The dataset separates human preferences into two categories: helpfulness and harmlessness. Helpfulness labels are used to train the reward model, while harmlessness labels are used to train the cost model. We utilize Gemini 2.5 Flash API [2] as an impartial judge to compute win rates, using consistent reward and cost models to evaluate all baselines. Experiments are based on the code [3] provided by Dai et al. (2024) and conducted on a server with NVIDIA A800x4 GPUs.

### D.2. Algorithms Implementation

We utilize the code [4] provided by the authors of RLSF (Reddy Chirra et al., 2024) for implementation. Other baselines including PPO-BT and PPO-Lag (Ray et al., 2019) are implemented based on the above code.

---

[1] https://huggingface.co/meta-llama/Llama-3.2-1B

[2] https://ai.google.dev/gemini-api

[3] https://github.com/PKU-Alignment/safe-rlhf

[4] https://github.com/shshnkreddy/RLSF

We implement the proposed PbCRL in Safety Gymnasium and autonomous driving environments based on the code [5]. The cost model $c_\psi$ is implemented as a 3-layer MLP with 64 hidden units per layer and ReLU activations. The policy $\pi_\theta$ is based on the PPO algorithm (Schulman et al., 2017). It uses an actor network to output a Gaussian distribution over actions. Two separate critic networks are used to estimate the reward value function ($V_\phi^R$) and the cost value function ($V_\varphi^C$). The actor network and both critic networks share the same architecture: a 4-layer MLP with 256 hidden units per layer and ReLU activations.

# E. Training Details

This section provides detailed procedures for training the proposed PbCRL algorithm, encompassing both the cost model and the policy. The cost model is trained using a two-stage training strategy: an offline pre-training phase followed by an online finetuning phase. The policy is trained by a policy gradient method guided by the learned cost model.

## E.1. Cost Model Training

Following standard PbRL protocols (Lee et al., 2021; Christiano et al., 2017), we employ a synthetic oracle based on ground-truth costs to simulate human preference feedback for Safety Gymnasium and driving tasks. This established practice ensures evaluation consistency and reproducibility at scale, especially in simulated environments where real-time human querying is prohibitively expensive. For language modeling tasks, we utilize a standard dataset with human-annotated preferences provided by (Dai et al., 2024).

In Safety Gymnasium and driving task, all methods are limited to the same total query budget of 20000 for constraint learning. We assign 18000 queries for offline pre-training and 2000 queries for online finetuning. The offline dataset is constructed by randomly sampling trajectories from interactions generated by a PPO-Lag agent, labels are then generated using the ground truth cost and threshold.

**Offline Pre-training** We train the cost model parameterized by $\psi$ using the proposed loss function $\mathcal{L}_{PbCI}$, including the pairwise preference loss $\mathcal{L}_{pair}$, safety loss $\mathcal{L}_{safe}^{DZ}$ and SNR loss $\mathcal{L}_{SNR}$:

$$\mathcal{L}_{pair}(\psi) = -\mathbb{E}_\mathcal{D}\Big[\mu_1 \log \sigma(C_\psi(\tau_2) - C_\psi(\tau_1)) + \mu_2 \log \sigma(C_\psi(\tau_1) - C_\psi(\tau_2))\Big]$$

$$\mathcal{L}_{safe}(\psi) = -\mathbb{E}_\mathcal{D}\Big[\epsilon \log \sigma(-C_\psi(\tau)) + (1-\epsilon) \log \sigma(C_\psi(\tau) - \delta)\Big]$$

$$\mathcal{L}_{SNR}(\psi) = -\zeta \frac{\mathrm{Var}(C_\psi(\tau))}{\mathcal{H}(p(\mu))}, \quad \text{where } \mathcal{H}(p(\mu)) = -\sum p(\mu) \ln p(\mu)$$

$$\mathcal{L}_{PbCI}(\psi) = \mathcal{L}_{pair}(\psi) + \mathcal{L}_{safe}^{DZ}(\psi) + \mathcal{L}_{SNR}(\psi) \tag{35}$$

During this phase, the dead zone parameter $\delta$ is fixed (e.g., $\delta = 1$) to ensure training stability. The cost model parameters are updated via gradient descent:

$$\psi \leftarrow \psi - lr_\psi \nabla \mathcal{L}_{PbCI}(\psi) \tag{36}$$

We adopt the Adam optimizer with a learning rate of $1e^{-4}$ for the cost model. The batch size is set to 512. This procedure leverages offline preference data without incurring costly online labeling, enabling the cost model to learn a general estimate of the cost function. Early stop is applied to avoid overfitting.

**Online Finetuning** The pre-trained cost model is then used for policy optimization. During policy training, we select a comparatively small amount of generated trajectories for online labeling, which are then used to finetune the cost model parameter $\psi$ using the same loss function.

Crucially, the dead zone parameter $\delta$, which was frozen during pre-training, is now adaptively updated to calibrate the safety boundary. Since the ground truth expectation $\mathbb{E}[C(\tau)]$ is inaccessible, we utilize the **violation rate** as a surrogate metric for alignment. We posit that the empirical violation probability $\mathbb{P}_{vio}$ (from labels) should align with the model's predicted

---

[5]https://github.com/hmhuy0/SIM-RL

violation probability $\hat{\mathbb{P}}_{vio}$. For a batch of online data $\mathcal{B}$, these probabilities are estimated as:

$$\hat{\mathbb{P}}_{vio} = \frac{1}{|\mathcal{B}|} \sum \mathbb{I}(c_\psi(\tau) > 0) \qquad \mathbb{P}_{vio} = \frac{1}{|\mathcal{B}|} \sum \mathbb{I}(\epsilon = 0) \tag{37}$$

The dead zone parameter $\delta$ is updated to minimize the discrepancy between these two rates:

$$\mathcal{L}_\delta = \|\mathbb{P}_{vio} - \hat{\mathbb{P}}_{vio}\|^2 \tag{38}$$
$$\delta \leftarrow \delta - lr_\delta \nabla_\delta \mathcal{L}_\delta \tag{39}$$

This adaptive mechanism ensures that the learned cost model maintains a consistent interpretation of safety standards as the policy evolves. By performing labeling at a much lower frequency than policy updates, this design significantly reduces the annotation burden compared to fully online methods like RLSF (Reddy Chirra et al., 2024).

### E.2. Policy Training

Policy optimization is performed using **PPO-Lag** (Ray et al., 2019), a standard approach for Safe RL. We define the policy network $\pi_\theta$, reward value network $V_\phi^R$, and cost value network $V_\varphi^C$.

The policy $\pi_\theta$ is trained to maximize reward while satisfying the learned constraint $\mathcal{J}^{C_\psi}(\pi) \leq 0$. The objective incorporates a Lagrange multiplier $\lambda$ into the advantage estimation. The policy loss is given by:

$$\mathcal{L}_\theta = -\mathbb{E}_{\pi_\theta} \left[ \min \left( r_t(\theta) A_t^{Lag}, \text{clip}(r_t(\theta), 1 - \epsilon_{clip}, 1 + \epsilon_{clip}) A_t^{Lag} \right) \right] \tag{40}$$

where $r_t(\theta) = \frac{\pi_\theta(a_t|s_t)}{\pi_{\text{old}}(a_t|s_t)}$ is the probability ratio. The Lagrangian advantage $A_t^{Lag}$ combines reward and cost advantages:

$$A_t^{Lag} = A_t^R - \lambda A_t^C \tag{41}$$

Here, $A_t^R$ and $A_t^C$ are calculated using Generalized Advantage Estimation (GAE) based on the reward $r(s, a)$ and the learned cost $c_\psi(s, a)$, respectively. The policy parameters are updated as:

$$\theta \leftarrow \theta - lr_\theta \nabla_\theta \mathcal{L}_\theta \tag{42}$$

The value networks are trained to minimize the mean squared error (MSE) against the respective returns:

$$\mathcal{L}_\phi = \frac{1}{2} \mathbb{E} \left[ (V_\phi^R(s) - R_{target})^2 \right], \quad \phi \leftarrow \phi - lr_\phi \nabla_\phi \mathcal{L}_\phi \tag{43}$$
$$\mathcal{L}_\varphi = \frac{1}{2} \mathbb{E} \left[ (V_\varphi^C(s) - C_{target})^2 \right], \quad \varphi \leftarrow \varphi - lr_\varphi \nabla_\varphi \mathcal{L}_\varphi \tag{44}$$

where $C_{target}$ is computed using the learned cost $c_\psi$.

Finally, the Lagrange multiplier $\lambda$ is updated via dual ascent to enforce the safety constraint:

$$\lambda \leftarrow \max \{0, \lambda + lr_\lambda \mathbb{E}_{\pi_\theta}[C_\psi]\} \tag{45}$$

### E.3. Detailed Pseudo-code

Algorithm 2 presents the complete workflow of PbCRL, explicitly detailing the parameter updates for the cost model ($\psi$), dead zone ($\delta$), policy ($\theta$), value functions ($\phi, \varphi$), and Lagrange multiplier ($\lambda$).

---

**Algorithm 2** Preference-based Constrained Reinforcement Learning (PbCRL)

---

**Require:** Preference dataset $\mathcal{D}$, fine-tune interval $K$
 1: **Phase 1: Cost Model Pre-training**
 2: Initialize cost model parameters $\psi$ and dead zone $\delta$
 3: **for** each epoch **do**
 4:     Sample a batch of preference pairs from $\mathcal{D}$
 5:     Update $\psi$ by minimizing $\mathcal{L}_{PbCI}$ (Eq. 36)
 6: **end for**
 7:
 8: **Phase 2: Policy Optimization**
 9: Initialize policy $\theta$, value networks $\phi, \varphi$, and multiplier $\lambda$
10: **for** iteration $n = 0, 1, \ldots, N$ **do**
11:     Collect trajectories using current policy $\pi_\theta$
12:     **if** $n \bmod K = 0$ **then**
13:         **// Online Finetuning**
14:         Sample a small batch of trajectories $\mathcal{B}$ for online labeling; update $\mathcal{D} \leftarrow \mathcal{D} \cup \mathcal{B}$
15:         Update dead zone $\delta$ using violation rate alignment (Eq. 39)
16:         **for** each fine-tuning epoch **do**
17:             Sample a batch from $\mathcal{D}$
18:             Update $\psi$ by minimizing $\mathcal{L}_{PbCI}$ (Eq. 36)
19:         **end for**
20:     **end if**
21:     **// Policy Update**
22:     Compute advantages $A^R$ and $A^C$ using learned cost $c_\psi$
23:     Update policy $\theta$ via PPO-Lag objective (Eq. 42)
24:     Update value networks $\phi, \varphi$ (Eq. 44)
25:     Update Lagrange multiplier $\lambda$ (Eq. 45)
26: **end for**

---

# F. Experiments

### F.1. Overall Results

We provide additional empirical results and extended analysis in this section. Figure 6 illustrates the training curves for average episode return (Row 1) and true cumulative safety cost (Row 2) across three navigation tasks. Additionally, Row 3 presents the average learned cost of the cost model for three algorithms: PbCRL, RLSF, and PPO-BT. The learned cost in PbCRL (blue) and PPO-BT (green) has threshold $\hat{d} = 0$, while RLSF (orange) assumes the ground truth threshold $d$ a priori and use it to constrain the learned cost. We observe that the true cost (Row 2) of all algorithms except RLSF converges towards a steady state at the end of training, indicating the policies satisfy their safety constraints, either learned (PbCRL, PPO-BT) or ground truth (PPO-Lag).

First, we consider PPO-Lag (red), optimized using the ground truth cost. It consistently serves as an effective upper bound, demonstrating high return while satisfying true constraints across tasks. In contrast, RLSF (orange) yields the lowest return and cost among all methods. This behavior is explained as an overestimation issue in its cost learning procedure from trajectory-level segments (Reddy Chirra et al., 2024). As shown in Figure 6 (Row 3), the learned costs are significantly high despite the true cost being low, resulting in an overly conservative learned constraint. As optimizing against an excessively tight constraint often sacrifices the potential return for safety, RLSF achieves suboptimal performance despite maintaining low cost.

Next, we compare our proposed PbCRL (blue) with PPO-BT (green). PbCRL consistently achieves high return performance comparable to the PPO-Lag upper bound, while successfully maintaining its true cost (Row 2, blue) slightly below the safety threshold (black dashed line). We observe that the learned cost of PbCRL (Row 3, blue) converges to $\hat{d} = 0$ as well. This indicates that by satisfying the learned constraint $\mathbb{E}[C_\psi(\tau)] < \hat{d} = 0$, the PbCRL policy is capable of meeting the true safety constraint $\mathbb{E}[C(\tau)] < d$. This finding, along with the balanced performance of PbCRL between return and safety, similar to PPO-Lag, provides strong evidence that our learned constraint effectively aligns with the genuine safety requirements.

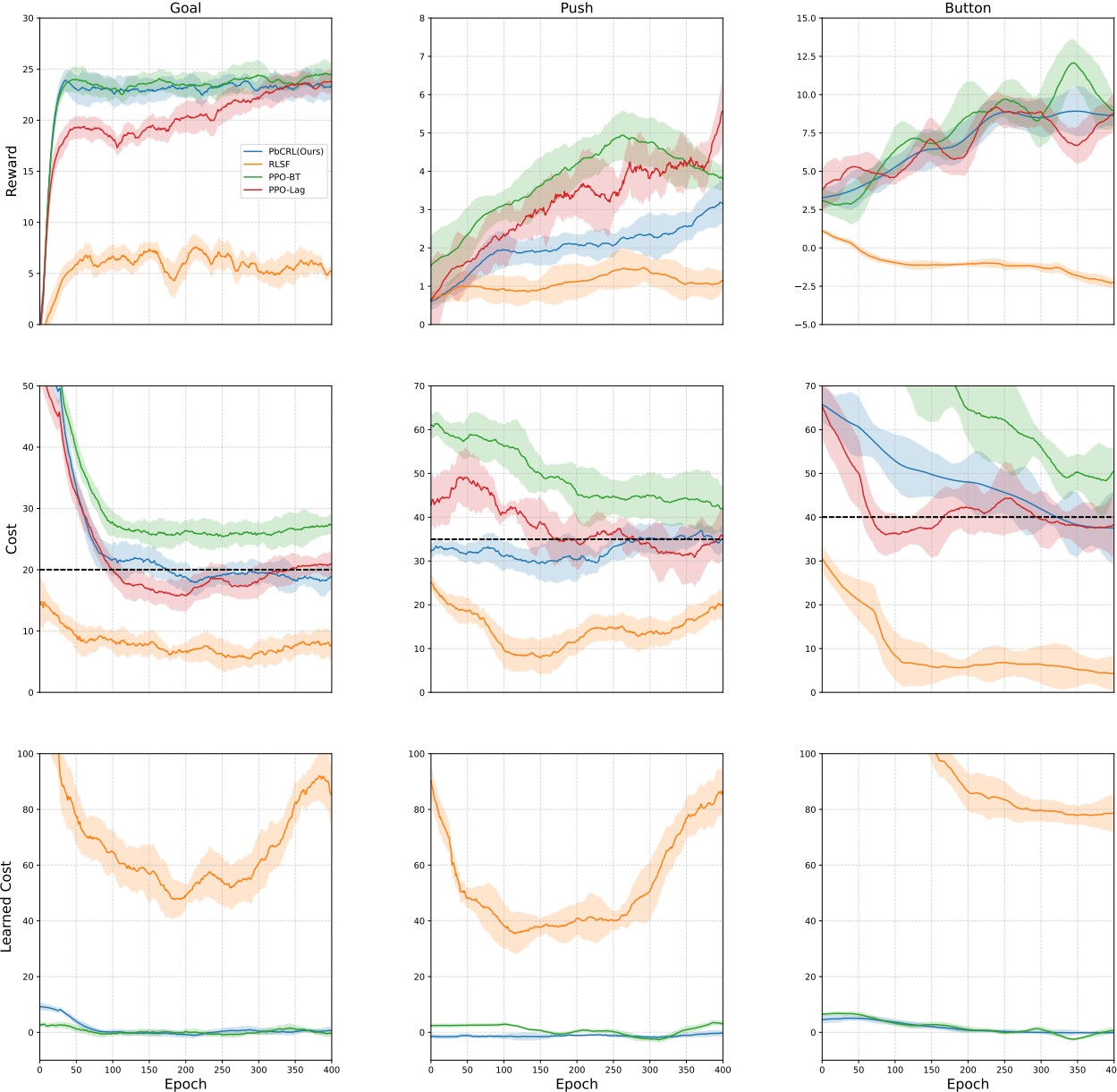

*Figure 6.* Average episode return (Row 1), cost (Row 2) and learned cost (Row 3) of four algorithms on Goal (left), Push (middle) and Button (right) tasks. Dashed lines are safety thresholds.

*Table 9.* Empirical results on three navigation tasks in Safety Gymnasium

| Tasks (Threshold) | Metrics | PPO-Lag (Oracle) | PbCRL (Ours) | RLSF | PPO-BT |
|---|---|---|---|---|---|
| Goal (20) | Return | $23.50 \pm 1.14$ | $\mathbf{23.41 \pm 1.08}$ | $5.78 \pm 1.07$ | $24.26 \pm 0.99$ |
| | Cost | $20.70 \pm 1.91$ | $\mathbf{18.50 \pm 2.03}$ | $8.20 \pm 2.21$ | $26.97 \pm 2.62$ |
| | Learned Cost | N/A | $0.50 \pm 1.26$ | $89.52 \pm 8.07$ | $\mathbf{0.12 \pm 1.36}$ |
| | Accuracy | N/A | $\mathbf{87\% \pm 2\%}$ | $83\% \pm 2\%$ | $85\% \pm 1\%$ |
| Push (35) | Return | $4.77 \pm 0.67$ | $\mathbf{3.20 \pm 0.38}$ | $1.07 \pm 0.35$ | $4.09 \pm 0.57$ |
| | Cost | $36.17 \pm 5.00$ | $\mathbf{34.86 \pm 3.13}$ | $20.05 \pm 3.56$ | $41.85 \pm 5.65$ |
| | Learned Cost | N/A | $\mathbf{-0.73 \pm 0.78}$ | $80.48 \pm 9.00$ | $1.62 \pm 0.46$ |
| | Accuracy | N/A | $\mathbf{86\% \pm 3\%}$ | $80\% \pm 2\%$ | $84\% \pm 2\%$ |
| Button (40) | Return | $8.96 \pm 1.06$ | $\mathbf{8.73 \pm 0.78}$ | $-2.01 \pm 0.37$ | $9.01 \pm 1.51$ |
| | Cost | $38.25 \pm 5.20$ | $\mathbf{37.51 \pm 5.65}$ | $4.26 \pm 3.33$ | $50.78 \pm 8.10$ |
| | Learned Cost | N/A | $\mathbf{-0.12 \pm 0.60}$ | $78.04 \pm 4.80$ | $-0.51 \pm 0.95$ |
| | Accuracy | N/A | $80\% \pm 3\%$ | $73\% \pm 3\%$ | $\mathbf{82\% \pm 2\%}$ |

Conversely, PPO-BT, despite also achieving high return comparable to PPO-Lag, fails to meet true safety requirement. We observe that the learned cost (Row 3, green) in PPO-BT converges to $\hat{d} = 0$, indicating the policy satisfies the learned constraint $\mathbb{E}[C_\psi(\tau)] < \hat{d} = 0$. However, the true cost (Row 2, green) exceeds the safety threshold (black dashed line), indicating the violation of the true safety constraint. This discrepancy suggests that the learned constraint in PPO-BT fails to accurately reflect the true constraint. As discussed in Section 3, we attribute this failure to the standard BT model's tendency, to induce a symmetric estimated cost distribution rather than a heavy-tailed one, shown in Figure 1. This leads to underestimation of the expected cost and consequently overly aggressive policies.

Table 9 provides a quantitative summary of the final performance metrics. PbCRL demonstrates superior overall performance by effectively balancing the trade-off between return and safety, outperforming other baselines. Moreover, confirming the visual trend in Figure 6, the true cost achieved by PbCRL policies consistently converges to values closely around the safety threshold, mirroring the behavior of PPO-Lag trained with the ground truth cost. This empirical finding validates that the constraint learned by PbCRL functions as a reliable surrogate for the true underlying constraint, highlighting the effectiveness of our constraint learning method.

### F.2. Cost Distribution Analysis

In above section, we presented the empirical results of algorithms in terms of return and cost. While these metrics provide crucial insights into an algorithm's overall effectiveness and ability to satisfy the true constraint, they do not directly reveal why certain methods succeed or fail in learning the constraint itself. To gain a deeper understanding of the underlying constraint inference capabilities of each approach, we conduct a detailed analysis of the characteristics of the learned cost distributions.

As highlighted in Section 1, Section 3, the distribution properties of the cost are fundamentally linked to the expectation-based constraint $\mathbb{E}[C] < d$. Especially in safety-critical scenarios, where costs often exhibit heavy-tailed. An accurate learned cost function that captures these characteristics may lead to reliably estimate the true expected cost. Failure to do so, as demonstrated by standard BT models which tend to infer symmetric distributions, can lead to inaccurate expectation estimates and constraint misalignment.

To quantitatively assess the dissimilarity between cost distributions, we utilize the 2-Wasserstein distance (W2 distance). W2 distance quantifies the minimum "effort" required to transform one distribution into the other, this property makes the W2 distance particularly well-suited for comparing distributions that may differ significantly in their shape, location (mean), and scale (variance), providing a more comprehensive metric than, for example, simply comparing means or using measures

*Table 10.* Evaluation metrics: W2 distance and Bias between converged cost and threshold

| Tasks | PbCRL (Ours) | | RLSF | | PPO-BT | |
|---|---|---|---|---|---|---|
| | W2 | Bias | W2 | Bias | W2 | Bias |
| Goal | **16.8** | **1.5** | 68.1 | 11.8 | 42.7 | 6.9 |
| Push | **20.3** | **0.1** | 82.4 | 14.9 | 36.5 | 6.8 |
| Button | **44.3** | **2.5** | 165.5 | 35.7 | 78.8 | 10.78 |

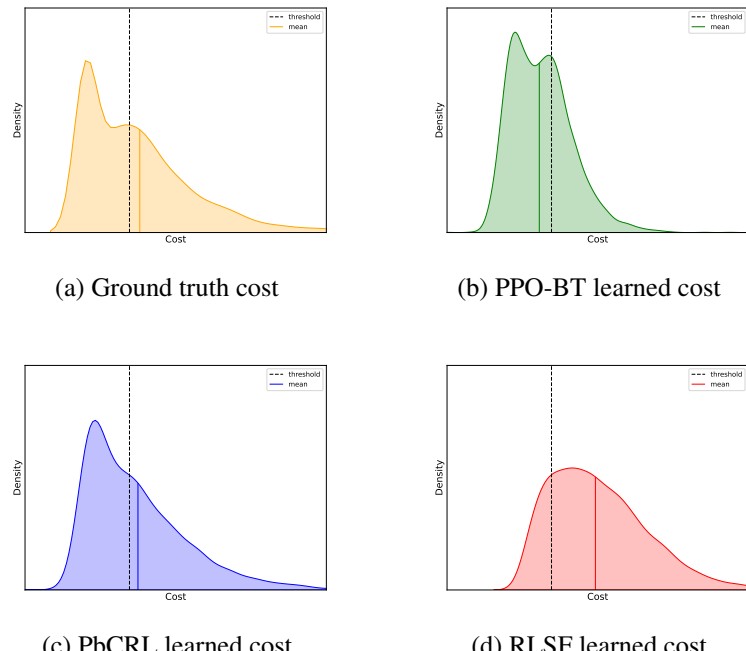

(a) Ground truth cost          (b) PPO-BT learned cost

(c) PbCRL learned cost          (d) RLSF learned cost

*Figure 7.* Cost distribution in Goal task. Solid line: expectation; Black dashed line: safety threshold.

like KL divergence which can be sensitive to differences in support or zero probabilities. A lower W2 distance indicates that the two distributions are closer to each other in terms of their overall structure.

Table 10 presents the W2 distances between the learned cost distributions (for PbCRL and the baselines) and the ground truth cost distribution across the three tasks. Analyzing these results, we observe that a lower W2 distance, signifying that a learned cost distribution is closer to the true heavy-tailed cost distribution, consistently correlates with improved constraint alignment and, as a consequence, better balanced policy performance in terms of safety and reward, as evidenced by the primary results presented in Table 9. Focusing on the performance of our proposed PbCRL, Table 10 shows that PbCRL consistently achieves the lowest W2 distances compared to other baselines across all tested task. These results provide quantitative evidence that the cost distribution learned by PbCRL is substantially closer to the true heavy-tailed safety cost distribution.

Furthermore, to gain a more intuitive and qualitative understanding of the learned cost functions and how they align with the true underlying safety costs, we sample 5000 trajectories and use cost models from PbCRL and the baselines to estimate the costs of these trajectories. The estimated costs are normalized, and their distributions are fitted using Kernel Density Estimation (KDE) and plotted in Figure 7. The solid vertical line represents the expectation of the respective cost distribution, and the dashed vertical line indicates the corresponding safety threshold.

The ground true cost distribution (a) is distinctly heavy-tailed. Its expectation (orange solid) exceeds the threshold (dashed), indicating constraint violation. PPO-BT (b) utilizes a standard BT model, resulting in a relatively symmetric cost distribution.

Despite achieving high accuracy (Table 9), this model underestimates the true expected cost. Its expectation (green solid) falls below the threshold (dashed), leading the agent to falsely believe the constraint is satisfied (Type II Error). RLSF learns a binary cost based on safe probability estimation. While achieving reasonable accuracy using its classification-based metrics (sum the logits of statewise safe probabilities to assess trajectory-level safety), the trajectory costs (d) by summing these binary costs shift towards higher values (red solid). This overestimation, explained as the performance degradation of RLSF when preference feedback is collected over longer segments such as trajectory level (Reddy Chirra et al., 2024), leads to overly conservative constraints and suboptimal return (Table 9).

In contrast to the standard BT model in PPO-BT and RLSF, the cost distribution learned by PbCRL (c) exhibits a heavy-tailed characteristic similar to the ground truth (a). Furthermore, the cost expectation of PbCRL (blue solid) behaves similarly to the ground truth cost expectation (orange solid), indicating better constraint alignment.

Above findings strongly suggest that the technical contributions in PbCRL, particularly the dead zone mechanism discussed in Section 3, are effective in encouraging the learned cost model to better approximate the heavy-tailed characteristic and structure of the true cost distribution, thereby facilitating superior constraint alignment and enabling the preferable policy performance, as observed in Figure 6 and Table 9

### F.3. Uniform cost threshold across tasks

As stated in the main text, cost thresholds are typically assigned task-specifically to scale with the difficulty of the task. This ensures that the evaluation remains informative and that the constraints are neither too loose nor too tight for any given task. To further validate this approach, we conduct experiments with a uniform cost threshold of 20 across all Safety Gymnasium tasks. As shown in Table 11, this uniform threshold causes performance collapse in Push and Button (returns fell to near zero) for all methods, rendering their capabilities indistinguishable.

*Table 11.* Performance Comparison with Uniform Cost Threshold (20) across Tasks

| Task | Metric | PPO-Lag (Oracle) | PbCRL (Ours) | RLSF | PPO-BT |
|------|--------|------------------|--------------|------|--------|
| HalfCheetah | Reward | 2803 | 2565 | 2316 | 2687 |
| | Cost | 20.78 | 20.25 | 14.52 | 22.50 |
| Walker2d | Reward | 3075 | 2790 | 2587 | 2853 |
| | Cost | 19.54 | 17.84 | 13.87 | 20.43 |
| Humanoid | Reward | 6248 | 5761 | 5483 | 6103 |
| | Cost | 19.68 | 19.20 | 17.27 | 21.84 |
| Goal | Reward | 23.50 | 23.41 | 5.78 | 24.26 |
| | Cost | 20.70 | 18.50 | 8.20 | 26.97 |
| Push | Reward | 1.81 | 1.77 | 0.73 | 1.90 |
| | Cost | 19.82 | 20.66 | 14.42 | 28.21 |
| Button | Reward | 2.62 | 1.78 | $-2.73$ | 1.97 |
| | Cost | 18.50 | 19.10 | 3.37 | 25.72 |

### F.4. Ablation Studies on Dead Zone parameter

We investigate the sensitivity of the dead zone parameter $\delta$ during the cost model pre-training phase, with results summarized in Table 12. Setting $\delta = 0$ reverts the framework to a standard BT model. As discussed in Section 3, this induces a symmetric cost distribution that fails to capture tail risks, resulting in a high W2 distance and systematic cost underestimation (leading to safety violations). A small dead zone ($\delta = 0.1$) offers only marginal improvements, insufficient to correct the distributional bias. Conversely, an excessively large dead zone ($\delta = 2$) over-corrects the distribution. While this ensures safety, it results in over-conservative policies (lower rewards) and an increased W2 distance due to severe distributional distortion. An appropriate dead zone parameter ($\delta = 1$) proves the best choice, achieving the lowest W2 distance. This confirms that a moderate dead zone effectively recovers the heavy-tailed characteristic of the true cost, striking the best balance between robust constraint satisfaction and reward maximization.

*Table 12.* Ablation studies on dead zone parameter $\delta$

| Task (Threshold) | Metrics | $\delta = 0$ (No) | $\delta = 0.1$ (Small) | $\delta = 1$ (Medium) (Ours) | $\delta = 2$ (Large) |
|---|---|---|---|---|---|
| Goal (20) | W2 distance | 38.9 | 33.5 | **16.8** | 29.1 |
| | Reward | $24.55 \pm 1.01$ | $24.79 \pm 1.18$ | $\mathbf{23.41 \pm 1.08}$ | $22.51 \pm 1.05$ |
| | Cost | $28.05 \pm 2.55$ | $27.46 \pm 1.89$ | $\mathbf{18.50 \pm 2.03}$ | $16.49 \pm 2.18$ |
| | Accuracy | $84\% \pm 2\%$ | $85\% \pm 2\%$ | $87\% \pm 3\%$ | $\mathbf{88\% \pm 2\%}$ |
| Push (35) | W2 distance | 36.1 | 35.8 | **20.3** | 23.6 |
| | Reward | $4.15 \pm 0.55$ | $3.81 \pm 0.21$ | $\mathbf{3.20 \pm 0.38}$ | $2.96 \pm 0.35$ |
| | Cost | $42.50 \pm 5.50$ | $40.26 \pm 2.15$ | $\mathbf{34.86 \pm 3.13}$ | $30.91 \pm 1.78$ |
| | Accuracy | $83\% \pm 3\%$ | $85\% \pm 1\%$ | $\mathbf{86\% \pm 3\%}$ | $86\% \pm 3\%$ |

## F.5. Ablation Studies on SNR-based Loss

$\mathcal{L}_{PbCI} = \mathcal{L}_{pair} + \mathcal{L}_{safe}^{DZ} + \mathcal{L}_{SNR}$ represents the total loss function used in the PbCRL framework. $\mathcal{L}_{pair}$ and $\mathcal{L}_{safe}^{DZ}$ are both cross-entropy losses operating on the same scale, thus they naturally take equal weights without requiring heuristic tuning. $\mathcal{L}_{SNR}$ (Section 4) is specifically designed to encourage the learned cost model to induce sufficient variance in its outputs, thereby providing more informative signals for efficient policy optimization. We conduct a sensitivity analysis on the weighting coefficient $\zeta$ of $\mathcal{L}_{SNR}$ in Goal task, summarizing mid-training and final performance in Table 13.

*Table 13.* Ablation study on SNR loss weight $\zeta$

| Metrics | $\zeta = 10^{-2}$ | $\zeta = 10^{-3}$ (Ours) | $\zeta = 10^{-5}$ | $\zeta = 0$ |
|---|---|---|---|---|
| Return (Mid) | $3.8 \pm 2.2$ | $\mathbf{22.9 \pm 1.5}$ | $22.5 \pm 1.3$ | $22.1 \pm 1.1$ |
| Cost (Mid) | $17.9 \pm 8.1$ | $\mathbf{18.4 \pm 2.3}$ | $24.5 \pm 2.5$ | $24.8 \pm 2.8$ |
| Return (End) | $6.5 \pm 1.8$ | $\mathbf{23.4 \pm 1.1}$ | $20.6 \pm 1.6$ | $20.7 \pm 1.4$ |
| Cost (End) | $21.9 \pm 5.9$ | $\mathbf{18.5 \pm 2.0}$ | $19.2 \pm 2.2$ | $18.4 \pm 2.6$ |

Results show that an excessive SNR weight ($10^{-2}$) destabilizes the learning process, evidenced by high variance in both return and cost. Conversely, removing (0) or under-weighting ($10^{-5}$) the SNR term leads to a flattened cost landscape, resulting in slow safety convergence (higher costs at mid-training). A balanced $\zeta$ ($10^{-3}$) induces sufficient cost differentiation, accelerating convergence with lower cost and higher return both at mid-training and final. This allows the algorithm to reach a desired performance level within less training iterations, particularly beneficial in real-world applications where interacting with the environment is expensive / time-consuming, or in computationally intensive domains like language model alignment where policy training is costly.

## F.6. Ablation Studies on Two-stage Training Strategy

We also evaluate the impact of the two-stage training strategy proposed in Section 5.1. We compare the full PbCRL (offline pre-training followed by adaptive online finetuning) against PbCRL-Offline (offline pre-training only) in the Goal and Push tasks, with training curves shown in Figure 8.

As previously discussed, relying solely on continuous online annotation is expensive and impractical. A primary advantage of our two-stage strategy, demonstrated by both PbCRL and PbCRL-Offline variants, is that they leverage offline dataset for the initial phase of cost model training. This design inherently reduces the burden of costly online labeling compared to purely online approaches. Furthermore, the performance achieved by PbCRL-Offline, which infers constraints solely from this offline data, yields policies with performance comparable to other baselines in Table 9.

The full PbCRL algorithm, which incorporates the adaptive online finetuning stage, consistently demonstrates further improvements in achieving better constraint satisfaction. Figure 8 (Column 2) illustrates that the cost of both methods converges, but the refinement offered by online finetuning enables the policy to achieve a cost (blue) that are closer to the

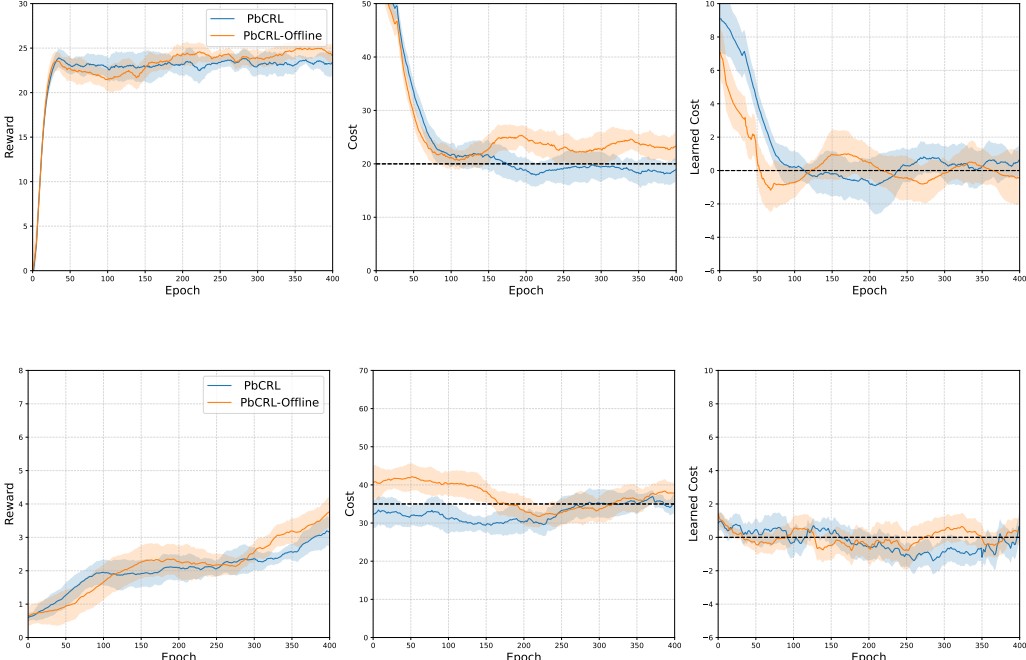

*Figure 8.* Ablation study on the two-stage training strategy in Goal (Row 1) and Push (Row 2) tasks.

safety threshold (dashed). This enhanced constraint satisfaction indicates that the adaptive online finetuning step, although utilizing a comparatively smaller amount of online annotation, effectively allows the learned cost model to better align with the genuine safety requirements of the environment where the agent is deployed.

### F.7. Robustness to Label Noise

To evaluate the resilience of our approach against imperfect human feedback, we conduct a sensitivity analysis under varying preference label noise levels ($\{0\%, 5\%, 10\%, 30\%\}$). This evaluation spans both the open-ended robotic navigation setting (*Goal*) and highly dynamic autonomous driving tasks (*Block*). The fidelity of the learned cost distribution is quantified using the 2-Wasserstein (W2) distance against the ground-truth cost distribution.

*Table 14.* W2 distance under different preference label noise levels in robotic (*Goal*) and driving (*Block*) tasks.

| Task | Noise Level | PbCRL (Ours) | PPO-BT |
|------|-------------|--------------|--------|
| **Goal** | 0% (Clean) | **16.8** | 42.7 |
| | 5% | **24.5** | 53.4 |
| | 10% | **36.8** | 60.1 |
| | 30% | 82.8 | **79.6** |
| **Block** | 0% (Clean) | **5.9** | 6.0 |
| | 5% | **6.9** | **6.9** |
| | 10% | 7.3 | **7.2** |
| | 30% | **9.4** | 10.2 |

Table 14 summarizes the W2 distances across different noise levels. As expected, introducing preference label noise leads to a natural degradation in constraint alignment (manifested as increased W2 distances) for both PbCRL and the standard BT baseline (PPO-BT). In the open-ended *Goal* task, PbCRL consistently maintains a lower W2 distance compared to PPO-BT across low to moderate noise levels (0%–10%). This suggests that our dead-zone mechanism provides an effective

tolerance buffer, preserving the heavy-tailed characteristics and structural alignment of the cost function even under moderate perturbations. At extreme noise levels (30%), both methods suffer more pronounced performance drops due to the highly unconstrained nature of the navigation plane.

In autonomous driving scenarios, the absolute W2 distances and the rate of degradation are considerably smaller than those in robotic navigation. This milder degradation is primarily because highway driving features highly structured safety boundaries—such as discrete lane lines and explicit vehicle proximity limits—which inherently constrain the cost distribution more tightly than in open-ended settings. Despite this different domain structure, PbCRL demonstrates resilience.

### F.8. Extended Analysis on Language Model Alignment

To provide a more comprehensive understanding of the language model (LLM) alignment experiments presented in Section 6.2, we discuss the performance characteristics of the baselines.

*Table 15.* Language Model Alignment Results

| Metric | PPO | RLSF | Safe RLHF | PbCRL (Ours) |
|---|---|---|---|---|
| Win Rate (Helpfulness) ↑ | 72.5% | 60.4% | 79.4% | **80.7**% |
| Win Rate (Harmlessness) ↑ | 60.7% | 54.2% | 76.5% | **82.1**% |
| Reward ↑ | 2.78 | 1.31 | 4.05 | **4.22** |
| Cost ↓ | $-0.57$ | 1.10 | $-2.97$ | **$-3.03$** |

As shown in Table 15, RLSF (Reddy Chirra et al., 2024) yields suboptimal performance. Originally designed for continuous control, RLSF estimates state-wise binary costs from segment-level feedback. Adapting it to sequence-level LLMs causes a methodological mismatch, yielding suboptimal performance.

In physical control tasks like Safety Gymnasium, reward (velocity) and cost (collisions) are intrinsically conflicting. However, in LLM alignment, helpfulness (reward) and harmlessness (cost) exhibit a unique semantic coupling (positive correlation). This coupling is empirically evidenced by our reward-only PPO baseline (Table 15), which inherently achieves a 60.7% harmlessness win rate over the SFT model even without any cost model. By recovering heavy tail (Dead Zone) and ensuring informative gradients (SNR), PbCRL accurately identifies safe-yet-helpful responses, shifting the entire Pareto frontier upward instead of merely trading off.

