# OpenReview forum: "Safe Reinforcement Learning with Preference-based Constraint Inference"
_ICML.cc/2026/Conference — ICML 2026 regular_

### Official Review · Reviewer_PzJ9 · 2026-03-08

**Soundness:** 3
**Presentation:** 4
**Significance:** 4
**Originality:** 3
**Overall Recommendation:** 5
**Confidence:** 4

**Summary:**

The paper presents Preference-based Constrained Reinforcement Learning (PbCRL), a novel framework to handle preferences that are not explicitly specified. These *human* preferences are inferred from pairwise comparisons. The authors identify that the standard Bradley-Terry (BT) preference models are incapable to approximating the hidden cost distribution (as in the case of human preference). especially when they are heavy tailed. This risks underestimation of the average cost leading to unsafe choices in downstream policies. To address this underestimation and overall bad *fit*, they proposed PbCRL which augments the standard BT model with a dead zone margin, signal to noise ratio (SNR) and two stage training strategy. The dead-zone margin which intuitively pushes the cost of unsafe trajectories towards the right helps in addressing the underestimation of the underlying cost distribution. The SNR loss aims to reduce the likelihood of a flattened cost distribution as it can hinder policy gradients downstream. The two stage training (pretraining and finetuning) aims to address the expensive labelling of preferences in real time. The framework utilizes existing offline preference data to train a cost model which is them deployed in an online environment. The framework is tested on Safety Gymnasium, an autonomous driving simulator and a LM alignment experiment.

**Compliance With Llm Reviewing Policy:**

Affirmed.

**Final Justification:**

The authors addressed all my questions in the review. The added experiments they performed further strengthen the paper. Their framework has wide applicability in RL and therefore I maintain my positive score.

**Key Questions For Authors:**

**Major Questions**
1. Based on the results in Table 1, is it possible to set the actual safety limit = safety limit * 1.5 (or 2), similar to a safety factor? PPO-BT achieves better rewards but crosses the safety limit by a margin. With the safety factor, it could stay within the actual safety limit even if it crosses the set safety limit? Would that obtain better rewards than PbCRL?
2. In figure 2, for Push and Button tasks, it seems that PPO-BT hasn't converged yet. Isn't it likely it would achieve better costs and quite possibly go below the cost limit?
3. In table 4, why are costs negative? Is it some sort of transformed loss like log?
4. The proof in Appendix C upper bounds the norm of the cost gradient, however, maximizing the upper bound does not necessarily correlate to having higher gradient norms. It would *allow* a higher gradient norm but its not necessary the higher norms would be achieved. Is there a mechanism that pushes the gradient norms to higher values? Additionally, high variance in the cost model could simply introduce gradient noise, which might destabilize the Lagrange multiplier $\lambda$ and lead to the instability seen ablation studies when $\zeta=10^{-2}$.
5. While the authors propose an adaptive calibration mechanism for $\delta$ based on matching violation rates (Eq. 14), Table 10 demonstrates that the W2 distance is highly sensitive to the initial or fixed values of $\delta$ used during pre-training. Since the violation rate is a coarse discrete signal based on indicator functions, can the authors provide a mathematical justification or empirical evidence showing that matching the frequency of violations (indicator function) is sufficient to recover the correct expectation of the cost distribution (the integral)?
6. Isn't it possible for modern optimization methods like Adam to offset the higher gradient norms given by the Dead zone mechanism? Wouldn't larger adaptive step sizes on the last layer gradient's achieve similar heavier right-tail inducing effects.

**Minor Points**
1. In table 1 caption, it would be clearer to write bold methods satisfy the set cost limit (i.e. rewards and costs are not being compared individually).
2. In sec 6.3, is the ablation study meant to refer to Table 5 as well as Table 9?

**Limitations:**

No.

Potential negative impacts haven't been elaborated, but I am sure they can include it in the additional page if selected.

Only one limitation has been written. I think the authors can elaborate a bit more on possible technical limitations. Scenarios where the cost distribution has multiple modes or has a heavy left tail might offer unique problems with the proposed framework?

**Strengths And Weaknesses:**

**Strengths**

**Significance**: The motivation to include human preference which are not always explicitly mentioned is highly significant. Several applications exist where human preferences are present and agents adhering to such preferences are likely to be adopted in large number of applications. The framework presented can be adopted by researchers from several safe RL subdomains.

Incorporating the downstream policy performance in their work is also useful as preference prediction by itself is not a complete framework.

**Originality**: The identification of the limitation of BT models and how it effects the modeling of the cost distribution is is novel.

**Soundness**: The dead zone and SNR loss functions are well motivated by the understanding of the BT model. The two stage training also reduce online labeling burden.

The paper experiments on multiple continuous control tasks in Safety Gymnasium as well as autonomous driving. Ablations on the SNR weight and dead zone margins also provide better understanding of their framework.

**Presentation**: The paper is presented well with ablations, questions they attempt to answer, tables etc.


**Weakness**

**Soundness**: PPO-BT, consistently achieves higher rewards but violates safety limits. However, as noted in the results, it appears PPO-BT has not fully converged in complex tasks like "Push" and "Button".

Furthermore, it remains unclear if simply lowering the safety threshold for PPO-BT would yield an equally good reward-cost Pareto frontier than the complex PbCRL framework. (Refer to Q1)

The justification for the SNR loss relies on an upper bound of the cost gradient norm (Eq. 37). However, maximizing an upper bound does not necessarily lead to higher actual gradient norms or improved "informativeness" of the gradient signal.

 **Significance**: The existence of offline preference datasets might be strong for some domains or applications. 18000 offline human preferences in an offline dataset can still be considered quite expensive, especially for new domains where such large datasets have not been annotated or require some degree of expertise from the human annotators. However, it is still a useful framework for several applications

---

> ### Author Rebuttal · Authors · 2026-03-30
>
> We thank the reviewers for their insightful feedback and address the concerns below.
>
> **1. Lowering the safety threshold for PPO-BT**
>
>
> We thank the reviewer for the suggestion. We evaluated PPO-BT on the *Goal* task ($d=20$) with various lower safety thresholds to simulate a "safety factor." Results are summarized below:
>
> | Metrics    | PbCRL (Ours) | PPO-BT (20) | PPO-BT (18) | PPO-BT (15) | PPO-BT (10) |
> | :--------- | :------: | :---------: | :---------: | :---------: | :---------: |
> | **Reward** |  **23.41**   |    24.26    |    23.86    |    23.15    |    18.27    |
> | **Cost**   |  **18.50**   |    26.97    |    24.08    |    20.82    |    14.02    |
>
> While lowering threshold enables PPO-BT to achieve actual costs closer to the safety limit, providing an improved trade-off compared to the baseline PPO-BT (20). However, manually modifying the safety limit may not be a viable solution in practice. In our preference-based setting, the algorithm lacks direct access to the environment's true safety threshold; instead, safety boundary is implicitly **'baked into'** the collected preference labels, rather than explicitly provided.
>
>
>
> **2. PPO-BT has not fully converged in "Push" and "Button"**
>
> We continued training PPO-BT from 400 to 600 epochs. The results are summarized below:
>
> | Tasks (Threshold) | Metrics | PbCRL (Ours) | PPO-BT (400) | PPO-BT (600) |
> | :------- | :-- | :-----: | :---: | :----------: |
> | **Push** (35)     | Reward  |     **3.20**     |     4.09     |     4.15     |
> | | Cost |  **34.86** |    41.85     |    39.72     |
> | **Button** (40)   | Reward  |     **8.73**     |     9.01     |    10.04     |
> |  | Cost | **37.51**|    50.78     |    48.45     |
>
> As shown, PPO-BT’s cost stays above the limit because its learned model incorrectly classifies the policy as safe. This evidence proves the issue is structural cost underestimation.
>
>
>
> **3. Why cost in table 4 are negative**
>
> In the LLM experiment, costs represent the **estimated values (logits)** from a shared cost model used to evaluate all algorithms, where more negative values indicate safer (more harmless) outcomes.
>
>
>
> **4. Pushing higher gradient norms and stability**
>
> We agree that maximizing an upper bound does not strictly guarantee an increase in the actual gradient norm. Our analysis in Appendix C serves as a qualitative justification: the SNR loss is primarily intended to prevent cost "flattening", rather than a formal proof of strict norm growth. Eq. 34 indicates that policy gradients are sensitive to cost differentiation. By promoting cost variance, the SNR loss tends to encourage higher differentiation, which may help sustain larger policy gradient magnitudes.
>
> To address potential instability from high variance, we use a balanced weighting coefficient $\zeta$ for SNR loss, while standard techniques like gradient clipping can be further applied.
>
>
>
> **5. Recovering cost expectation from violation rates**
>
> We thank the reviewer for this insightful question. Empirically, Figure 8 demonstrates that online calibration via violation rates successfully aligns the learned cost closer to the threshold. Intuitively, we hypothesize this coarse matching is effective because the dead zone loss establishes a heavy-tailed structural prior during pre-training. The violation rate then acts as a anchor to calibrate its location.
>
> We acknowledge the indicator is a coarse signal, yet directly matching expected costs is impractical. Collecting absolute scalar cost labels online is prohibitively expensive, and estimating continuous expectations from small online batches likewise suffers from high sampling variance. Conversely, violation rates are trivially computable from the binary safety labels already present in the preference dataset.
>
>
>
> **6. Adam optimization and larger step size settings**
>
> On Adam offsetting the higher gradients: Empirically, we tracked the `last_layer_update_norm` (the actual parameter update magnitude). In early training, the dead zone yields a significantly higher update norm than standard BT, providing the necessary "initial push" to unsafe costs. Near convergence, even as update norms equalize, the final learned cost mean for unsafe trajectories is **3.0 (dead zone) vs. 1.9 (standard BT)**, while safe costs remain comparable. This confirms that Adam does not cancel out the impact of the dead zone.
>
> Regarding larger step sizes on the last layer, simply increasing adaptive step sizes for the last layer acts as a symmetric scalar multiplier. This would stretch the entire cost distribution—pushing unsafe costs higher while simultaneously pushing safe costs lower. This global scaling cannot resolve the symmetry bias of BT models. In contrast, sample-conditional mechanism like the Dead Zone can independently shift the mode of unsafe trajectories to recover the heavy-tailed profile without distorting the safe cost region.
>
> We thank the reviewer for the minor suggestions and will address them in the revision.

---

> > ### Author Rebuttal · Reviewer_PzJ9 · 2026-04-03
> >
> > Thank you for the detailed rebuttal and the additional experiments on the PPO-BT and continued training. I believe the extra PPO-BT experiments with lower safety thresholds should definitely be part of the main text as it strengthens your framework even more. The proposed framework holds potential to affect multiple RL sub fields that aim to learn some implicit ranking.
> >
> > All my questions have been adequately answered, and I maintain my positive score.

---

> > > ### Author Response · Authors · 2026-04-04
> > >
> > > We thank the reviewer for the positive feedback and support. As suggested, the additional experiments will be integrated into the main text to further strengthen the manuscript. Your recognition of our framework’s potential impact is greatly valued.

---

### Official Review · Reviewer_3MeP · 2026-03-08

**Soundness:** 3
**Presentation:** 3
**Significance:** 2
**Originality:** 3
**Overall Recommendation:** 4
**Confidence:** 3

**Summary:**

This work addresses the core challenge in Safe Reinforcement Learning (Safe RL) where real-world safety constraints are complex, implicit, and difficult to explicitly specify. It proposes the Preference-based Constrained Reinforcement Learning (PbCRL) framework, which introduces three key enhancements: a dead zone mechanism to induce heavy-tailed cost distributions (alleviating risk underestimation), a Signal-to-Noise Ratio (SNR) loss to increase cost variance (providing informative signals for effective policy learning), and a two-stage training strategy (offline pre-training + online fine-tuning) that reduces the burden of online labeling. The framework conducts experiments across three scenarios—Safety Gymnasium (6 tasks), autonomous driving (2 tasks), and language model alignment—comparing against baseline methods such as RLSF and PPO-BT.

**Compliance With Llm Reviewing Policy:**

Affirmed.

**Final Justification:**

Based on the authors’ thorough rebuttal and supplementary materials addressing issues including unified cost thresholds, hyperparameter and loss ablation studies, and offline-online training stability, I maintain my original recommendation of weak accept. Although some concerns remain unresolved and inconsistencies exist in the experimental results, the work is technically solid and valuable to the field, so the initial weak accept judgment is unchanged.

**Key Questions For Authors:**

1. Could the authors add comparisons with several existing safe reinforcement learning algorithms?
2. Is it possible to set the same cost threshold for different tasks, and conduct comparative experiments under multiple groups of different cost thresholds?
3. Is the dead-zone upper-bound parameter δ a hyperparameter? Could some necessary ablation experiments be added?
4. Are $L_{pair}+L^{DZ}_{safe}+L_{SNR}$ directly summed with equal weights? Could corresponding ablation experiments be supplemented?
5. Are there specific techniques adopted to stabilize the transition between offline training and online fine-tuning? How is training stability guaranteed?

**Limitations:**

We encourage the authors to elaborate on the limitations and operational boundaries of their method.

**Strengths And Weaknesses:**

Strengths
1. The paper provides necessary theoretical derivations and proofs (detailed derivations are not yet presented, so the correctness of the theory cannot be verified).
2. The paper is well-structured with clear and logical reasoning.


Weaknesses
1. The experimental validation appears insufficient. It is suggested to compare with other safe reinforcement learning algorithms, such as the projection method, dual method, and other related approaches.
2. In addition, why are the cost thresholds set differently across experiments? For instance, the cost thresholds for tasks such as goal, push, and button seem to be 20 and 40, respectively.
3. Necessary ablation studies are missing, such as those on the upper-bound parameter δ of the dead zone.
4. Some parameters in the model lack sufficient explanation and analysis. For example, are $L_{pair}+L^{DZ}_{safe}+L_{SNR}$ directly summed with equal weights? Furthermore, ablation experiments on this component are absent.
5. How is the stable transition achieved between the offline pre-training and online fine-tuning stages?

---

> ### Author Rebuttal · Authors · 2026-03-30
>
> We thank the reviewers for their insightful feedback and address the concerns below.
>
> **1. Add comparisons with several existing safe RL algorithms**
>
> We respectfully clarify that our work focuses on **constraint inference** (learning an unknown cost function $\hat{C}$ from preferences) rather than **safe policy optimization** (updating a policy using a known ground-truth cost $C$). Standard Safe RL algorithms (e.g., projection methods) require explicit ground-truth cost signals, which are unavailable in our preference-based setting.
>
> To rigorously evaluate the quality of the inferred constraints, we adopted a **controlled-variable design**. We employed PPO-Lagrangian as the unified policy optimization backbone for all evaluated methods (PbCRL, RLSF, PPO-BT, and the Oracle). By fixing the RL optimizer, we isolate the performance differences solely to the **fidelity of the learned cost models**. This ensures a fair comparison of constraint inference capabilities, without confounding them with the choice of downstream policy optimizers.
>
>
>
> **2. Is it possible to set the same cost threshold for different tasks**
>
> Scaling safety thresholds with task difficulty is a standard protocol in Safe RL [1] to ensure informative evaluation. We apologize if our discussion in Sec 6.1 and App D.1 was not sufficiently prominent. Uniform thresholds are often uninformative: loose limits render simple tasks vacuous, while tight limits make complex tasks infeasible.
>
> Experiments with a uniform threshold of 20 confirmed this: causing **performance collapse** in *Push* and *Button* (returns fell to near zero) for all methods, rendering their capabilities indistinguishable.
>
> | Task            | Metric | PPO-Lag (Oracle) | PbCRL (Ours) | RLSF  | PPO-BT |
> | :-------------- | :----: | :--------------: | :----------: | :---: | :----: |
> | **HalfCheetah** |   R    |       2803       |     2565     | 2316  |  2687  |
> |                 |   C    |      20.78       |    20.25     | 14.52 | 22.50  |
> | **Walker2d**    |   R    |       3075       |     2790     | 2587  |  2853  |
> |                 |   C    |      19.54       |    17.84     | 13.87 | 20.43  |
> | **Humanoid**    |   R    |       6248       |     5761     | 5483  |  6103  |
> |                 |   C    |      19.68       |    19.20     | 17.27 | 21.84  |
> | **Goal**        |   R    |      23.50       |    23.41     | 5.78  | 24.26  |
> |                 |   C    |      20.70       |    18.50     | 8.20  | 26.97  |
> | **Push**        |   R    |       1.81       |     1.77     | 0.73  |  1.90  |
> |                 |   C    |      19.82       |    20.66     | 14.42 | 28.21  |
> | **Button**      |   R    |       2.62       |     1.78     | -2.73 |  1.97  |
> |                 |   C    |      18.50       |    19.10     | 3.37  | 25.72  |
>
> **3. Ablation experiments on the dead-zone parameter $\delta$**
>
> Ablations for  $\delta \in \{0, 0.1, 1, 2\}$ were included in Appendix F.5. A moderate value $\delta=1$ consistently achieves the best alignment and reward-safety trade-off. We will move this to the main text.
>
>
>
> **4. Ablation experiments on the weights of $L_{pair}, L_{safe}^{DZ}, L_{SNR}$**
>
> $L_{pair}$ and $L_{safe}^{DZ}$ are both cross-entropy losses (log-probabilities) operating on the exact **same scale**, thus they naturally take equal weights without requiring heuristic tuning. Conversely, $L_{SNR}$ is scaled by a weighting coefficient $\zeta$. A comprehensive ablation study on $\zeta \in \{0, 10^{-5}, 10^{-3}, 10^{-2}\}$ was provided in **Appendix F.3 (Table 9)**, showing $\zeta=10^{-3}$ achieves balanced performance.
>
>
>
> **5. Specific techniques to stabilize the transition between offline and online training**
>
> Stability is ensured by: 1) Sparse update frequency via the fine-tuning interval K (Algorithm 1), which enforces a timescale separation and ensures gradual cost model evolution; and 2) Adaptive calibration (Eq. 14), which smoothly adjusts the dead-zone parameter δ based on online violation feedback to handle distribution shifts without destabilizing the optimization.
>
>
>
> [1] Gu, Shangding, et al. "Enhancing efficiency of safe reinforcement learning via sample manipulation." NeurIPS 2024

---

> > ### Author Rebuttal · Reviewer_3MeP · 2026-04-04
> >
> > We thank the authors for their detailed and patient response, as well as their dedicated efforts in advancing research in the field of safe reinforcement learning.
> > However, we still have significant concerns regarding two key aspects: the validity of the experimental evaluations presented in this work, and the lack of clear performance improvements demonstrated. Specifically, the curves shown in Figure 2 suggest that the proposed method’s future return performance on tasks such as Goal, Push, and Button appears to be inferior to that of the baseline algorithms, which is inconsistent with the results provided in Table 1.
> >
> > After carefully considering the authors’ feedback, we remain inclined toward a weak reject recommendation for this submission, but we choose not to revise our review score at this stage.

---

> > > ### Author Response · Authors · 2026-04-04
> > >
> > > We sincerely thank the reviewer for the continued engagement and patient review. We realize that our initial presentation and captions may have led to critical misunderstandings regarding experimental evaluations and performance improvement. We respectfully provide clarifications below, supported by extended experimental data.
> > >
> > > **1. Validity of the experimental evaluations**
> > >
> > > Regarding your observation on "future return performance," we agree that the reward curves in Figure 2 show upward trends. However, we respectfully clarify that there is no numerical inconsistency with Table 1: the data in Table 1 presents the **final performance at epoch 400 in Figure 2**, representing the results at the end of training, whereas your observation refers to potential trends beyond that scope.
> > >
> > > To address your concerns regarding the significance of these trends and the validity of our evaluation, we wish to clarify two fundamental aspects of our experimental evaluations:
> > >
> > > - **Reward vs. Safety:** In safety-critical applications, satisfying safety constraints is a **strict prerequisite**, not a soft trade-off. An unsafe policy is fundamentally unusable in real-world scenarios, rendering its marginally higher reward meaningless. Therefore, evaluating Safe RL algorithms often prioritizes constraint satisfaction over raw returns.
> > >
> > > - **Comparing future return against baselines in Figure 2:** We suspect the reviewer is comparing PbCRL (blue) against **PPO-Lag (red) and PPO-BT (green)**. Regarding PPO-Lag (red), we must emphasize that **PPO-Lag is an Oracle reference, NOT a competing baseline**. It is trained using the *ground-truth cost*, which is unavailable in a preference-based constraint inference setting. Its reward represents an ideal upper bound that PbCRL aims to approach, rather than surpass.
> > >
> > >   **For PPO-BT (green)**, to address your concern regarding the future performance in Figure 2, we extended the training of the most competitive baseline (PPO-BT) and our method from 400 to 600 epochs to ensure full convergence.
> > >
> > > | Tasks (Threshold) | Metrics | PbCRL (400) | PbCRL (600) | PPO-BT (400) | PPO-BT (600) |
> > > | :---------------- | :------ | :---------: | :---------: | :----------: | :----------: |
> > > | **Goal** (20)        | Reward  |    23.41    |  **23.78**  |    24.26     |    24.58     |
> > > |                   | Cost    |    18.50    |  **18.47**  |    26.97     |    26.76     |
> > > | **Push** (35)     | Reward  |    3.20     |  **3.92**   |     4.09     |     4.15     |
> > > |                   | Cost    |    34.86    |  **34.94**  |    41.85     |    39.72     |
> > > | **Button** (40)   | Reward  |    8.73     |  **9.05**   |     9.01     |    10.04     |
> > > |                   | Cost    |    37.51    |  **37.98**  |    50.78     |    48.45     |
> > >
> > > The extended results confirm that while PPO-BT’s reward continues to rise slightly, its **true cost plateaus significantly above the safety limit** (e.g., 48.45 vs. a threshold of 40 in *Button*). Because it consistently violates safety constraints, its higher reward gains are impractical for safety-critical deployment. The violation occurs because PPO-BT’s cost model incorrectly classifies the policy as safe, confirming that the failure is due to a **structural cost underestimation** inherent in the standard BT model. Our dead-zone mechanism fundamentally resolves this issue, enabling PbCRL to strictly satisfy safety thresholds across all tasks. Furthermore, PbCRL **stabilizes its cost much faster** because our SNR loss explicitly maintains cost variance, providing highly informative gradients for more efficient policy learning.
> > >
> > > **2. What Constitutes a "Performance Improvement" in Safe RL**
> > >
> > > Guided by the "Safety-First" principle, where satisfying safety constraints is a **strict prerequisite** before evaluating rewards, we summarize the performance of the three preference-based inference approaches to highlight our improvements:
> > >
> > > *   **PPO-BT** achieves slightly higher rewards, but consistently **violates the safety constraints** with costs significantly exceeding the thresholds. In real-world safety-critical systems (e.g., autonomous driving), an unsafe policy is fundamentally unusable, rendering its marginally higher reward meaningless.
> > > *   **RLSF** stays within safety limits but is overly conservative, leading to collapsed reward performance.
> > > *   **PbCRL (Ours)** achieves **a clear Pareto improvement** over existing baselines: By accurately aligning the constraint, PbCRL maximizes rewards *strictly subject to* satisfying the safety constraint. It provides a balanced performance that mirrors the Oracle (PPO-Lag), representing the most significant and usable performance gain in the context of preference-based Safe RL.
> > >
> > > We hope these clarifications and the additional experimental data fully address your concerns. We deeply appreciate your time and constructive feedback.

---

### Official Review · Reviewer_bp4A · 2026-03-10

**Soundness:** 3
**Presentation:** 2
**Significance:** 3
**Originality:** 3
**Overall Recommendation:** 5
**Confidence:** 4

**Summary:**

The manuscript tackles the problem of learning safety constraints for safe RL from human preference data. The authors propose Preference‑based Constrained Reinforcement Learning (PbCRL), a two‑stage framework that first pre‑trains a cost model on an offline preference dataset and then fine‑tunes it online while simultaneously optimizing a policy via a Lagrangian‑based safe RL algorithm (essentially PPO‑Lag). A central technical contribution is the introduction of a dead‑zone loss that pushes estimated costs for unsafe trajectories above a positive margin, thereby encouraging a heavy‑tailed cost distribution that better matches the true safety constraint. In addition, a signal-to-noise ratio penalty is added to increase the variance of the learned cost, which the authors argue improves the quality of the policy gradient signal. Experimental evaluation spans three domains---locomotion and navigation tasks from Safety Gymnasium, autonomous‑driving highway scenarios, and a language‑model alignment setting---and compares PbCRL against several baselines (PPO‑Lag with ground‑truth cost, PPO‑BT, RLSF, Safe‑RLHF). Results show that PbCRL achieves a favorable trade‑off between return and safety, with learned costs that closely align with the true safety threshold and lower Wasserstein distances to the ground‑truth cost distribution.

**Compliance With Llm Reviewing Policy:**

Affirmed.

**Final Justification:**

The paper offers a technically sound and original contribution to safe reinforcement learning by introducing a dead‑zone loss and an SNR‑based variance incentive that shape a heavy‑tailed cost distribution, and the authors demonstrate its practical impact across robotics, autonomous driving, and language‑model alignment. Their rebuttal fully resolves my primary concerns---providing clear guidelines for selecting delta via risk‑aware heuristics or Bayesian optimization, supplying additional noise‑robustness results for driving tasks, and reporting a modest wall‑clock speed‑up---while the remaining presentation issues are minor and will be polished. Accordingly, my overall recommendation remains a weak accept.

**Key Questions For Authors:**

- Table 10 shows results for several delta values, but the selection of delta = 1 as "optimal" appears task‑specific. Could the authors provide guidelines or an automated procedure for choosing delta in new domains?
- The paper reports Wasserstein distances under different noise levels for the Goal task. How does performance degrade in the autonomous‑driving setting, where label noise may be higher due to ambiguous human judgments?
- The offline pre‑training uses 18 k preference queries and the online stage 2 k. How does total wall‑clock time compare to fully online baselines when measured on the same hardware?
- How do the authors plan to integrate security concerns---e.g., adversarial attacks---into their framework?

**Limitations:**

yes

**Strengths And Weaknesses:**

The paper is technically sound: the presented lemmas and theorems convincingly argue that the dead‑zone loss yields strictly higher costs for unsafe trajectories and a heavier right tail in the cost distribution, and the proofs rely on standard smoothness assumptions common in safe‑RL analysis. Algorithmically, the two‑stage pipeline reduces annotation cost while allowing the constraint to adapt to the evolving policy, and the use of a Lagrange multiplier follows established safe‑RL practice. Empirically, the authors evaluate on diverse benchmarks with multiple seeds, reporting both return and cost as well as the Wasserstein distance between learned and true cost distributions, directly measuring constraint alignment. The main novelty lies in the dead‑zone loss together with an SNR‑based variance incentive, which together shift the focus from merely fitting a pointwise cost to matching a heavy‑tailed cost distribution---an angle not previously explored in constraint inference. Presentation is generally clear, but many equations appear as stand‑alone displays rather than being embedded in sentences, which disrupts the narrative flow; notation is occasionally overloaded, and some figures lack clear legends. In terms of significance, learning safety constraints from cheap preference data addresses a practical bottleneck for deploying safe RL in real‑world systems where hand‑crafted constraints are infeasible, and the demonstrated near‑oracle safety performance with modest labeling budgets could influence robotics, autonomous driving, and language‑model alignment.

---

> ### Author Rebuttal · Authors · 2026-03-30
>
> We thank the reviewers for their insightful feedback and address the concerns below.
>
> **1. Guidelines for choosing delta in new domains**
>
> We thank the reviewer for this insightful question and suggest a principled pipeline for new domains: As a general guideline, the initial $\delta$ should reflect the environment's **risk profile**: domains prone to severe cascading effects (i.e., low-probability, catastrophically high-cost scenarios) require a larger $\delta$ to enforce a stronger heavy-tailed prior, whereas milder environments necessitate smaller values.
>
> To automate this selection, the tuning process can be formulated as a black-box optimization problem. For instance, using Bayesian Optimization (BO), one can evaluate a few initial $\delta$ values against a task-specific safety metric (e.g., collision rate). The BO algorithm can then model the $\delta \to \text{safety}$ mapping to propose the optimal $\delta$ with minimal environment rollouts. Furthermore, precise initialization is not strictly necessary. As detailed in Sec 5.1, our adaptive online fine-tuning dynamically calibrates $\delta$ during training, inherently enhancing constraint alignment.
>
>
>
> **2. Wasserstein distances under different noise levels in driving setting**
>
> We thank the reviewer for this suggestion. We have evaluated the W2 distances under varying label noise levels in the driving tasks:
>
> | Task      | Noise Level | PbCRL (Ours) | PPO-BT  |
> | :-------- | :---------- | :----------: | :-----: |
> | **Block** | 0% (Clean)  |   **5.9**    |   6.0   |
> |           | 5%          |     6.9      |   6.9   |
> |           | 10%         |     7.3      | **7.2** |
> |           | 30%         |   **9.4**    |  10.2   |
> | **Lane**  | 0% (Clean)  |   **7.3**    |   8.5   |
> |           | 5%          |     9.6      | **9.2** |
> |           | 10%         |   **10.5**   |  11.6   |
> |           | 30%         |   **12.9**   |  13.5   |
>
> As expected, increasing noise gradually degrades constraint alignment for both methods. However, compared to robotic navigation, the absolute W2 distances and the performance drops across noise levels are smaller. This milder degradation is likely because highway driving entails highly **structured safety boundaries** (e.g., discrete lanes and explicit proximity limits), which inherently constrain the cost distribution more tightly than open-ended robotic tasks.
>
>
>
> **3. Wall‑clock time of two-stage training and fully online baselines**
>
> PbCRL requires 5 hours in total (1.5h offline pre-training + 3.5h online), whereas fully online methods take 6 hours. Processing 18k queries offline enables rapid, large-batch convergence, bypassing the bottleneck of sequential online querying. Beyond saving wall-clock time, this two-stage design effectively lower the need for continuous annotation during policy training.
>
>
>
> **4. Plan to integrate security concerns**
>
> We thank the reviewer for raising this perspective. Integrating defenses against adversarial attacks is indeed a vital future direction. While our label noise study in App. F.6 provides a preliminary verification of the model's robustness to data perturbations, we plan to explicitly incorporate adversarial training moving forward. By augmenting the preference dataset with worst-case trajectory pairs, we can further strengthen and harden the inferred safety boundaries.

---

> > ### Author Rebuttal · Reviewer_bp4A · 2026-04-01
> >
> > The authors have adequately addressed all of the substantive concerns raised in my review.
> >
> > Guidelines for choosing delta:
> > They propose a clear, risk‑aware heuristic, describe an automated Bayesian‑optimization pipeline, and note that the adaptive online fine‑tuning step further mitigates sensitivity to the initial value. This directly addresses my request for guidance on selecting delta in new domains.
> >
> > Noise robustness in the driving setting:
> > New experiments on Wasserstein distances under varying label‑noise levels are provided, showing that PbCRL remains competitive with PPO‑BT and that degradation is modest in structured highway environments. This satisfies the request for additional robustness evidence.
> >
> > Computational overhead:
> > They report concrete wall‑clock times (5 h total vs. 6 h for fully online baselines) and explain how the offline pre‑training reduces the need for sequential querying, addressing my efficiency question.
> >
> > Security concerns:
> > Although not required for the current paper, the authors acknowledge the limitation and outline a future direction (adversarial training), showing awareness of broader safety issues.
> >
> > These responses resolve the technical questions that informed my confidence and recommendation. The only remaining shortcoming is the presentation issue, which was noted in the original review but is unrelated to the rebuttal content. Consequently, the overall recommendation and confidence scores remain unchanged.

---

> > > ### Author Response · Authors · 2026-04-04
> > >
> > > We thank the reviewer for the insightful feedback. We will polish the presentation of the paper as suggested.

---

### Official Review · Reviewer_K8kS · 2026-03-10

**Soundness:** 2
**Presentation:** 2
**Significance:** 2
**Originality:** 2
**Overall Recommendation:** 4
**Confidence:** 4

**Summary:**

This paper studies the challenge of learning complex and implicit safety constraints in Safe RL when explicit constraint specification or expert demonstrations are unavailable. The authors propose Preference-based Constrained Reinforcement Learning (PbCRL), a framework that infers safety costs from human trajectory preferences. They identify that the commonly used BT preference model fails to capture the asymmetric and heavy-tailed nature of safety costs, leading to risk underestimation. To address this issue, PbCRL contains a dead-zone mechanism in the preference model to better represent heavy-tailed cost distributions, an SNR-based loss to encourage informative exploration through cost variance, and a two-stage training strategy that combines offline preference learning with adaptive online fine-tuning to reduce labeling effort. Experiments show that PbCRL improves alignment with true safety constraints and achieves better safety and reward performance than baselines in both low-dimension tasks, such as robot navigation and autonomous driving, and high-dimension space such as LLM post-training.

**Compliance With Llm Reviewing Policy:**

Affirmed.

**Final Justification:**

The rebuttal has addressed my main concerns.

**Key Questions For Authors:**

(1) Confusing teaser figure. In Figure 1, the authors provide a comparison involving the proposed method. However, the figure requires additional explanation. In particular, the authors should clarify why the proposed method has advantages over the baseline and explain the definitions and terminology used in the figure, such as Type II error, to make the presentation more self-contained.

(2) Motivation for varying constraint thresholds. What is the motivation for setting different constraint thresholds across tasks? Could the authors report the full results for thresholds (5,20,35,40) for the tasks presented in Table 1?

(3) Inconsistent behavior of Safe-RLHF. In the language model alignment setting, Safe-RLHF (PPO-BT) appears to exhibit more conservative behavior, with lower reward but safer outcomes. However, in the Safety-Gymnasium and autonomous driving tasks, Safe-RLHF is reported to achieve both higher reward and higher cost, suggesting more aggressive behavior. Could the authors discuss this discrepancy in more detail?

(4) Human annotation process. Could the authors clarify how the human annotators provided preference labels (mentioned in lines 143–144) for the Safety-Gymnasium tasks and the autonomous driving tasks?

**Limitations:**

Yes

**Strengths And Weaknesses:**

Strength:

(1) Important and interesting problem. Safety in decision-making and safe reinforcement learning is both an important and practically relevant research problem.

(2) Broad-domain evaluation. The authors conduct experiments across several domains, including robotic locomotion, autonomous driving, and language models. Although the experiments on language models are not very comprehensive, they still demonstrate the potential of the method to extend to broader domains.

Weakness:

(1) Incremental novelty. The core component of this work is the so-called Dead Zone, which is defined by relaxing the safety requirement using 𝛿 δ. While this design aligns with intuition and shows improvements over the baselines, the corresponding theoretical analysis in Lemma 3.1 and Theorem 3.2 appears relatively straightforward and somewhat trivial.

(2) Missing ablation study. The authors should discuss the role of 𝛿 δ in the experimental section and conduct ablation experiments with varying values of 𝛿 δ to analyze how performance changes with this parameter.

(3) Missing baseline. In the language model alignment experiment, the baseline RLSF is omitted. Could the authors provide results for this baseline for a more complete comparison?

(4) Writing clarification. Although the authors successfully convey the main idea, several important points remain unclear and require further clarification. Please see the question section for more detailed comments.

---

> ### Author Rebuttal · Authors · 2026-03-30
>
> We thank the reviewers for their insightful feedback and address the concerns below.
>
> **1. Incremental novelty**
>
> We respectfully clarify a misunderstanding: the Dead Zone does not "relax" safety requirements. While standard BT learns $\hat{C}>0$  for unsafe trajectories, our mechanism forces stricter with $\hat{C}>$ 𝛿 (𝛿>0). This acts as a **structural shaper** to recover the true heavy-tailed cost distribution, mitigating risk underestimation.
>
> Theoretically, Lemma 3.1 and Theorem 3.2 are not isolated claims but essential stepping stones for **Corollary 3.3**. Lemma 3.1 establishes gradient amplification, which Theorem 3.2 extends via functional gradient descent (Appendix B) to prove strict dominance during multistep optimization. Corollary 3.3 links these micro-dynamics to macro-statistics, proving First-Order Stochastic Dominance to rigorously confirm **heavy-tail recovery**. We appreciate your feedback and will restructure these parts to make logical chain clearer.
>
> **2. Missing ablation study on 𝛿**
>
> Ablations on varying 𝛿 were included in Appendix F.5. A moderate 𝛿 achieves balanced performance. We will move this to the main text.
>
> **3. Missing baseline in LLM alignment**
>
> We implemented and evaluated RLSF in the LLM alignment task. Updated Table 4 results:
>
> | Metric | PPO | RLSF | Safe RLHF | PbCRL (Ours) |
> |:-|:-:|:-:|:-:|:-:|
> | Win Rate (Helpfulness) ↑ | 72.5% | 60.4% | 79.4% | **80.7%** |
> | Win Rate (Harmlessness) ↑ | 60.7% | 54.2% | 76.5% | **82.1%** |
> | Reward ↑ | 2.78 | 1.31 | 4.05 | **4.22** |
> | Cost ↓ | −0.57 | 1.10 | −2.97 | **−3.03** |
>
> Originally designed for continuous control, RLSF estimates state-wise binary costs from segment-level feedback. Adapting it to sequence-level LLMs causes a methodological mismatch, yielding suboptimal performance.
>
> **4. Confusion in Figure 1**
>
> Figure 1 plots KDE-fitted PDFs of cumulative costs for identical trajectories under three models. Dashed and solid lines denote the safety threshold and expectation $\mathbb{E}[C]$, respectively.
>
> (a) Ground Truth: heavy-tailed. $\mathbb{E}[C]$ exceeds the threshold, indicating constraint violation.
>
> (b) Standard BT: A symmetric distribution, causing $\mathbb{E}[C]$ to fall below the threshold. Type II Error in statistics means failing to reject a false null hypothesis. In our setting, it means **dangerously misclassifying unsafe violations** as 'safe'.
>
> (c) Ours: The dead zone recovers the heavy-tail. The estimated $\mathbb{E}[C]$ exceeds the threshold, correctly identifying the violation to match ground-truth.
>
> We will update the figure caption to be self-contained. Further empirical analysis on cost distributions is provided in Appendix F.2.
>
> **5. Motivation for varying constraint thresholds**
>
> Scaling safety thresholds with task difficulty is a standard protocol in Safe RL [1] to ensure informative evaluation. We apologize if our discussion in Sec 6.1 and Appendix D.1 was not sufficiently prominent. Uniform thresholds are often uninformative: loose limits render simple tasks vacuous, while tight limits make complex tasks infeasible.
>
> Experiments with a uniform threshold of 20 confirmed this: causing **performance collapse** in *Push* and *Button* (returns fell to near zero) for all methods, rendering their capabilities indistinguishable.
>
> |Task|Metric|PPO-Lag (Oracle)|PbCRL (Ours)|RLSF|PPO-BT|
> |:-|:-:|:-:|:-:|:-:|:-:|
> |**HalfCheetah**|R|2803|2565|2316|2687|
> ||C|20.78|20.25|14.52|22.50|
> |**Walker2d**|R|3075|2790|2587|2853|
> ||C|19.54|17.84|13.87|20.43|
> |**Humanoid**|R|6248|5761|5483|6103|
> ||C|19.68|19.20|17.27|21.84|
> |**Goal**|R|23.50|23.41|5.78|24.26|
> ||C|20.70|18.50|8.20|26.97|
> |**Push**|R|1.81|1.77|0.73|1.90|
> ||C|19.82|20.66|14.42|28.21|
> |**Button**|R|2.62|1.78|-2.73|1.97|
> ||C|18.50|19.10|3.37|25.72|
>
> **6. Inconsistent behavior of Safe-RLHF**
>
> We politely clarify a misunderstanding in LLM alignment, higher win rate (harmlessness) and lower cost indicate safer. PbCRL dominates Safe-RLHF in LLMs on both reward (4.22 vs 4.05) and cost (-3.03 vs -2.97).
>
> Unlike the speed-collision conflict in robotics, LLM harmlessness and helpfulness are **semantically coupled**. This is empirically evidenced by the reward-only PPO baseline, which inherently achieves a 60.7% harmlessness win rate over the SFT model even without any cost model. By recovering heavy tail (Dead Zone) and ensuring informative gradients (SNR), PbCRL accurately identifies safe-yet-helpful responses, shifting the entire Pareto frontier upward instead of merely trading off.
>
> **7. Human annotation process**
>
> Following standard preference-based RL [2], we used a scripted teacher based on ground-truth costs to simulate preferences for gym and driving tasks. This ensures consistency and reproducibility.
>
> [1] Gu, Shangding, et al. "Enhancing efficiency of safe reinforcement learning via sample manipulation." NeurIPS 2024
>
> [2] Christiano, Paul F., et al. Deep reinforcement learning from human preferences. NeurIPS 2017.

---

> > ### Author Rebuttal · Reviewer_K8kS · 2026-04-02
> >
> > I thank the authors for their rebuttal and have increased my score accordingly.

---

> > > ### Author Response · Authors · 2026-04-04
> > >
> > > We thank the reviewer for the positive feedback and for increasing the score. We will incorporate the discussed clarifications and additional results into the next version of the paper.

---

### Decision · Program_Chairs · 2026-04-30

**Decision:**

Accept (regular)

**Comment:**

The authors propose a safe RL algorithm with constraint inference using a dead-zone mechanism and exploration using a SNR loss. Experiments on a variety of benchmarks including robotics, autonomous driving and LLM alignment were used to validate the proposed algorithm. The contribution is interesting as it extends beyond the popular BT model to cover heavy tails.

The reviewers found the paper's contributions interesting and scored it positively on the acceptance side. Moreover, several technical queries were addressed during the rebuttal. Hence, I'm going with an acceptance recommendation.